# Longitudinal analysis of body weight reveals homeostatic and adaptive traits linked to lifespan in diversity outbred mice

G. V. Prateek [1], Zhenghao Chen[1], Kevin Wright[1,3], Andrea Di Francesco [1], Vladimir Jojic[1,4], Gary A. Churchill [2] & Anil Raj [1] ✉

Dense temporal measurements of physiological health, using simple and consistent assays, are essential to characterize biological processes associated with aging and evaluate the effectiveness of interventions on these processes. We measured body weight in 960 genetically diverse female mice, every 7-10 days over the full course of their lifespan. We used a state space model to characterize the trajectories of body weight throughout life and derived novel traits capturing the dynamics of body weight, 10 of which were both heritable and associated with lifespan. Genetic mapping of these body weight-derived traits identified 5 genomic loci, none of which were previously mapped to body weight. We observed that the ability to maintain stable body weight, despite fluctuations in energy intake and expenditure, was positively associated with lifespan in an age-dependent manner and mapped to a genomic locus linked to energy homeostasis. Our results highlight how dense longitudinal measurements of physiological phenotypes offer new insights into the biology of aging.

Aging is characterized by the progressive loss of homeostasis – the ability of an organism to maintain physiological integrity in response to intrinsic and extrinsic changes[1–5]. Dense longitudinal phenotyping throughout the lifespan of an organism enables us to measure homeostasis by quantifying temporal relationships in one or more phenotypes at multiple time-scales, and associate these measures to healthspan[6] and lifespan[7,8]. Among the broad array of phenotypes typically measured in aging studies, body weight serves as a particularly accessible and integrative phenotype, reflecting the combined effects of genetics, behavior, environment, metabolic activity, and disease processes throughout life.

Diet and metabolic activity are known to play central roles in modulating body weight, body composition[9], skeletal health[10], and aging[11,12]. Overconsumption of calories increases the risk of obesity and metabolic diseases[13], whereas dietary interventions such as caloric restriction (CR) and intermittent fasting (IF) have been shown to improve health and extend lifespan in diverse organisms[8,14–16]. While these interventions have been known to reduce inflammation, enhance metabolic efficiency, and promote cellular repair, how these effects on diverse biological processes mediate subsequent lifespan extensions remain poorly understood.

To address these questions, we conducted a longitudinal study (DRiDO: Dietary Restriction in Diversity Outbred Mice) to investigate the effects of dietary intervention on health and lifespan in genetically diverse female mice[8]. In addition to the comprehensive battery of health assessments performed every 6 months to 1 year, the study also measured the body weight of each mouse every seven to ten days from weaning until death, providing the opportunity to explore temporal fluctuations in a metabolic phenotype and its impact on health and lifespan. Previous research has associated changes in body weight over time with metabolic health, reproductive cycles, seasonal changes, disease susceptibility, and survival[17–19]. In this study, we leveraged the longitudinal body weight trajectories from the DRiDO study to investigate the impact of diet and genetics on short-term and long-term

[1]Calico Life Sciences LLC, South San Francisco, CA, USA. [2]The Jackson Labs, Bar Harbor, ME, USA. [3]Present address: Actio Biosciences, San Diego, CA, USA. [4]Present address: Kymeral LLC, San Francisco, CA, USA. ✉e-mail: anil@calicolabs.com

dynamics of body weight, and their subsequent effects on lifespan. Using these time-series body weight data from 960 genetically diverse female mice, we derived novel phenotypes that go beyond static measures to describe dynamic processes such as weight stability, fluctuations, and recovery from stress. These phenotypes allowed us to ask specific questions: How do diet and genetics influence body weight homeostasis and dynamics as mice age? Do phenotypes derived from short-term fluctuations in body weight predict lifespan? How much does genetics influence these derived traits?

In this study, we observed that the ability to maintain body weight was associated with increased lifespan in an age-dependent manner. Furthermore, CR and IF diets appeared to train mice for broader adaptation to stress, extending beyond diet-induced stress, as evidenced by their sustained ability to recover from extrinsic stressors throughout their lifespan. Finally, we mapped several genetic loci to these body weight-derived traits, none of which overlapped with genetic loci previously mapped to body weight itself[20]. These findings highlight that dense longitudinal measurements and the resulting dynamics of body weight provide new insights into the biology of aging, revealing how genetics and environment shape health and longevity.

## Results

### Outline of study design

We entered a total of 960 diversity outbred (DO) female mice derived from eight inbred founder strains (A/J, C57BL/6J, 129S1/SvImJ, NOD/ShiLtJ, NZO/HILtJ, CAST/EiJ, PWK/PhJ, and WSB/EiJ)[21] across six generations of breeding into the study[8], and measured their body weight every 7–10 days over the full course of their lifespan (Methods). From each generation, we obtained two batches of mice, resulting in 12 cohorts in the overall study. From enrollment (3 weeks) until 6 months of age, all mice were fed ad libitum (AL) on a diet of standard rodent chow (5KOG LabDiet). At 6 months of age, mice were randomized by housing cage to one of five dietary interventions (192 mice per group): ad libitum feeding (AL), 20% calorie restriction (20), 40% calorie restriction (40), fasting 1 day per week (1D), and fasting 2 consecutive days per week (2D) (Fig. 1A). Mice from each of the 12 cohorts were randomized across these five treatment groups.

For mice on intermittent fasting (IF) diets, fasting was imposed every week on Wednesday at 15:00, for 24 or 48 h for the 1D and 2D groups, respectively. These mice had unlimited food access (similar to AL mice) on their non-fasting days (Fig. 1B). For mice on the calorie restriction (CR) diets, a measured amount of food was provided daily at around 15:00; 2.75 grams/mouse/day for 20% CR and 2.06 grams/mouse/day for 40% CR. On Friday around 15:00, CR mice were given three times the amount of food until the following Monday afternoon (Fig. 1B). The amount of food provided to CR mice was determined based on a previous internal study at the Jackson Laboratory where the average amount of food consumed by female DO mice at 6 months of age was estimated to be 3.43 grams/day[8].

Mice in the DRiDO study underwent three cycles of health assessments at early, middle, and late life. These assessments included a 7-day metabolic cage run at around 4, 15, and 27 months of age; blood collection for flow cytometry analysis at around 5, 17, and 29 months; rotarod, body composition, echocardiogram, acoustic startle, bladder function, free wheel running, and a blood collection for complete blood count at around 10, 22, and 34 months of age. Furthermore, manual frailty and grip strength assessments were carried out at 6 month intervals (Supplementary Fig. 1A).

### Modeling the dynamics of body weight

Following the onset of dietary intervention, diet was the largest source of variation in body weight, among shared factors in this study (Fig. 1C, Supplementary Fig. 1B). Within each diet group, we observed additional variation in body weight across mice and ages (Fig. 1D); however

generation (and associated confounders such as month of enrollment, season of the year, etc) contributed little to variation in the body weight trajectories (Supplementary Fig. 1C). Some of the short-term variation in body weight could be attributed to the phenotyping tests to which the mice are subjected (Fig. 1C). For example, at 10, 24, and 35 months of age, all mice undergo nine tests over a period of one month, which involve handling by a technician or a change in their environment. In order to jointly model both the short-term and long-term temporal variation in body weight, we developed an auto-regressive hidden Markov model (ARHMM) that captures the dynamics of body weight using discrete latent states (Fig. 2A).

The ARHMM combines an autoregressive model and a hidden Markov model to represent a body weight trace (time-series) as a sequence of latent states. These latent states correspond to distinct patterns in the body weight that are shared across all mice. They do not directly model the biological processes that determine a mouse's body weight (e.g., metabolic function, estrus cycle, activity levels, food consumption, etc). The latent states evolve according to a Markov process, capturing the switching dynamics of body weight states over time. The autoregressive model captures the relationship between the current and past observations conditional on the current latent state (Methods). We trained the ARHMM with 70% of body weight traces from each dietary group. To determine the number of latent states of the ARHMM, we computed the deviance information criteria[22] on the held-out body weight traces for different numbers of latent states and starting from different random initializations (Supplementary Fig. 2A). We obtained the smallest deviance information criteria for a model of order $K = 3$. For the given model order, the random initialization which resulted in the highest evidence-based lower bound (ELBO) on the training set was selected as the best model (Supplementary Fig. 2B–D). Based on the sign and magnitude of the intercept parameters for the three latent states ($\phi_0^1$, $\phi_0^2$, and $\phi_0^3$), we named the states as decline, steady, and growth (Supplementary Table 1).

For each body weight trace, we used the best-fitting ARHMM model to infer, at each time point, the posterior probability of the mouse belonging to one of the three physiological states. The state with the highest posterior probability was then assigned to that time point (Fig. 2B). We found little overlap in the inter-quartile range of estimated rates of change of body weight within each state, indicating that these states can be well-resolved throughout the lifespan of a mouse (Fig. 2C). Our inferences under the ARHMM model allowed us to derive several state-related traits, such as state occupancy and state transitions. For every mouse, the state occupancy was defined as the percentage of time the mouse spent in each state (Fig. 2D). Similarly, the state transition was defined as the frequency with which the mouse switched from one state to another (Fig. 2E). We accounted for uncertainties in state assignments when deriving these traits by using a weighted average approach, where the posterior probabilities of the state assignments were used as weights. In the next sections, we explore the effects of age, diet, and genetics on these traits and how informative they are in predicting mouse lifespan.

### Calorie restriction improves body weight homeostasis throughout life

Homeostasis is the maintenance of physiological processes in response to intrinsic and extrinsic changes experienced by an organism[23]. Body weight homeostasis is defined as a state of unchanging body weight in response to environmental perturbations[24]. Modeling the dynamics of body weight allows us to generalize the definition of body weight homeostasis to include maintenance of constant body weight dynamics. We hypothesize that homeostasis of body weight dynamics can be influenced by a combination of various factors including diet, genetics, energy imbalance, and environmental stress.

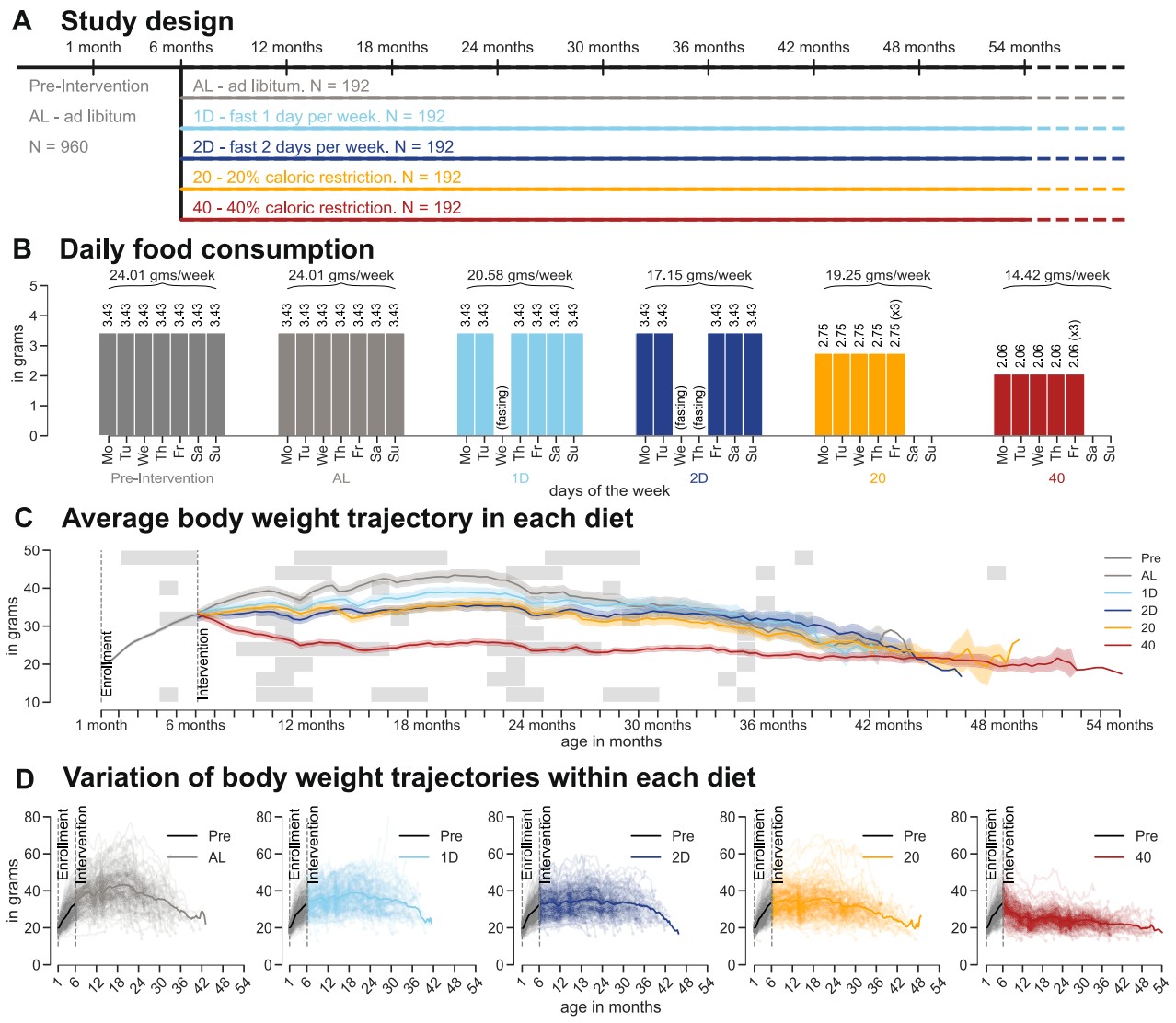

**Fig. 1 | Overview of the study design and data. A** At six months age, mice were randomly and equally divided across five dietary groups: ad libitum (AL), one-day fasting (1D), two-day fasting (2D), 20% calorie restricted (20), and 40% calorie restricted (40). Prior to dietary intervention, all mice were on ad libitum diet. **B** Average daily food consumption (in grams) per day of the week in the pre- and post-intervention phases across diet groups. Mice belonging to the 20% and 40% groups were given three times (3x) their daily average intake every Friday. **C** Average body weight (solid lines) and standard errors (shaded region) under different dietary interventions. Each horizontal gray bar represents a phenotyping event and the width of the gray bar captures the mean ± standard deviation of the duration (in months) of the phenotyping event when 95% of mice were assayed. **D** Body weight trajectories of mice in each dietary group. The colored solid line represents the average body weight trajectory of mice in each diet.

To characterize homeostasis of body weight dynamics across diets as mice age, we divided the time-axis into 6-month non-overlapping intervals starting from birth to 42 months. For each age bin, we defined steady state homeostasis as the fraction of time spent in the steady state (Fig. 3A, left panel). We observed that, on average, mice spent over 60% of time post-intervention in steady state homeostasis. However, we observed large variation in steady state homeostasis across diets; on average, over a 6-month period, 40% CR mice could maintain body weight for nearly 4.5 months while AL mice could do so for just 3.5 months. When we changed the time-axis from 6-month interval bins to ten percent interval bins of proportion of life lived, we observed that 40% CR mice spent nearly 75% of their time in steady state homeostasis into the last decile of life (Fig. 3B, left panel). We could redefine steady state homeostasis in terms of the empirical probability of not transitioning out of steady state. The higher this probability, the more stable the steady state is in the mouse. We observed similar trends in the effects of age and diet on steady state

homeostasis when it was redefined in terms of state stability (Supplementary Fig. 3A–B).

Similar to steady state homeostasis, we defined growth state homeostasis as the fraction of time spent in the growth state (Fig. 3A, center panel) and decline state homeostasis as the fraction of time spent in the decline state (Fig. 3A, right panel). As expected, we observed that AL mice, on average, gained body weight for at least 4 months out of their first 6 months of life; this growth state homeostasis dropped to approximately 20% at 18 months of age. Unsurprisingly, CR and IF significantly reduced the ability of a mouse to sustain growth, with 40% CR mice spending less than 3 weeks gaining body weight over the year following dietary intervention. On the other hand, we observed that decline state homeostasis in AL mice substantially increased with age after 18 months, with mice spending nearly 30% of their time on average losing body weight until the end of their life. In contrast, calorie restriction severely reduced decline state homeostasis, with 40% CR mice spending only 10–15% of their time losing

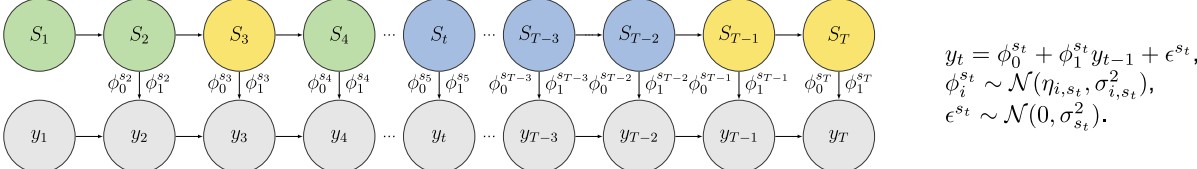

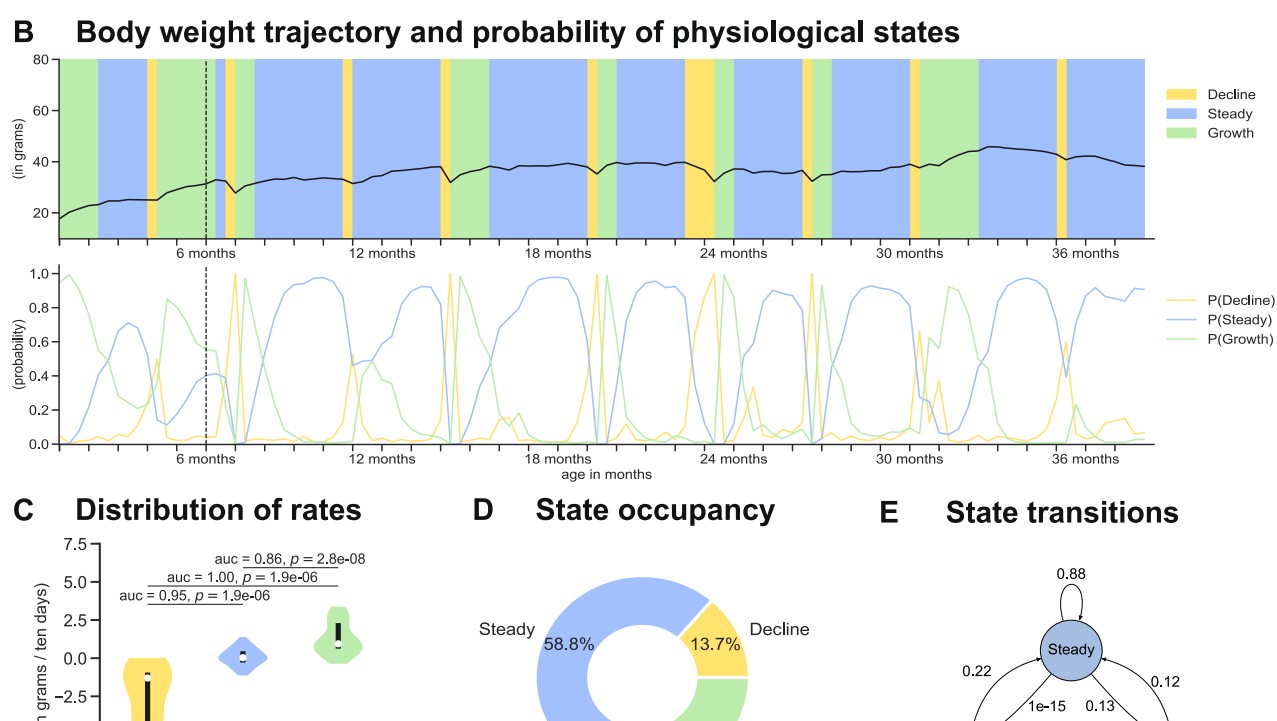

**Fig. 2 | ARHMM and body weight-derived traits for a sample mouse. A** A graphical illustration of an autoregressive hidden Markov model (ARHMM). Here, $y_{1:T}$ are body weight measurements of a mouse and $S_{1:T}$ are the corresponding latent states specified by the ARHMM. **B** Sample body weight trace of a mouse on a one-day fasting diet. The start time of the intervention is indicated with a dashed vertical line. The underlying states (growth, steady, and decline) are represented using background colors (top panel). The posterior probability trace for each latent state for the corresponding body weight trace (bottom panel). **C** Distribution of the model parameters in each latent state. To assess the statistical significance between the parameters of any two given latent states, a two-sided Mann–Whitney U test was performed, and the corresponding $p$-values were calculated. The solid black line and white dot represent the interquartile range and the median, respectively. **D** The percentage of time spent in each latent state. **E** Empirical state transition probabilities computed using the inferred states for the sample mouse.

body weight after 18 months. Similar trends were also observed in the effects of age and diet on growth and decline state homeostasis when the time-axis was changed to proportion of life lived and homeostasis was defined in terms of state stability rather than state occupancy (Supplementary Fig. 3C–F).

We quantified age-independent summaries of body weight trajectories post-intervention such as longest continuous bout and maximum rate of change of body weight conditioned on the latent state, and explored the effects of diet on these measures. Calorie restriction significantly increased the longest continuous bout in steady state while IF had no significant effect on this trait (Fig. 3C, left panel). The average longest stretch of maintaining body weight lasted longer than 30% of the post-intervention lifespan for the average CR mouse compared to approximately 25% for the average AL mouse. (Supplementary Fig. 4A, left panel). On the other hand, both CR and IF significantly shortened the longest continuous bout of gaining or losing body weight (Fig. 3C, Supplementary Fig. 4A, center and right panels). While both CR and IF affected the lifespan-normalized time of

onset of the longest stretch of body weight gain, only CR affected the lifespan-normalized time of onset of the longest stretch of body weight loss (Supplementary Fig. 4B–C, center and right panels).

Steady state homeostasis is disrupted when mice experience a sudden change in body weight. To quantify this disruption, we calculated the maximum rate of change of body weight in the growth and decline states. AL mice achieved significantly faster post-intervention gain and loss of body weight compared to mice on other diets (Fig. 3D). However, the effects of diet on these phenotypes were no longer significant when the rates were normalized by body weight, measured at the time when the maximum rates were recorded (Supplementary Fig. 4D). Nonetheless, the average mouse could achieve an 8% per week increase in body weight and a 15% per week drop in body weight, suggesting that stressful life events can cause strong perturbations to body weight. However, diet did not have a substantial effect on the time at which the maximum rate of change in body weight were recorded (Supplementary Fig. 4E–F).

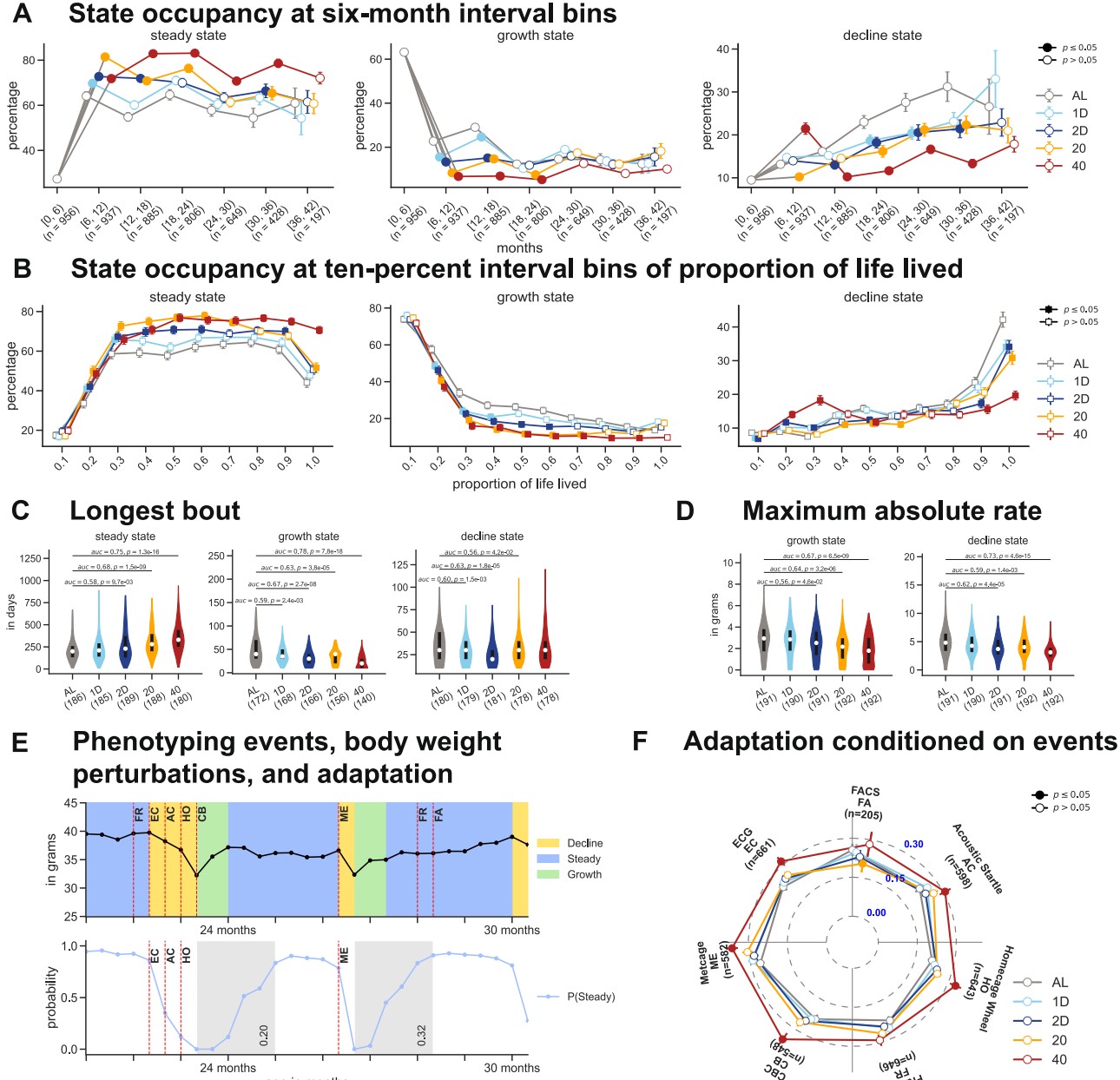

**Fig. 3 | Influence of diet and age on body weight homeostasis. A** The mean (+/−
s.e.) state occupancy in steady, growth, and decline states, at the 6-month interval
bins. **B** The mean (+/− s.e.) state occupancy in steady, growth, and decline states, at
ten percent interval bins of proportion of life lived. **C** The longest continuous bout
in the growth, steady, and decline states in the post-intervention phase. **D** The
maximum absolute rate of growth and decline of body weight in the post-
intervention phase. **E** A zoomed-in plot of the body weight trace of a sample mouse
on one-day fasting diet. The age of the mouse at the phenotyping event is indicated
using red-dashed line (top panel). The posterior probability of steady state for the

corresponding body weight trace (bottom panel). The time period of adaptation to
stress are indicated using gray boxes. **F** Radar chart of the average value and the
standard-error of adaptation to stress conditioned on the phenotypic assay.
In (**A**), (**B**), and (**F**), solid squares or circles indicate $p < 0.05$, where the $p$-values
were obtained by performing a two-sided Mann–Whitney test between a diet
group and the AL group conditioned on the same interval bin or phenotyping assay.
In (**C**) and (**D**), $p$-values were obtained by performing a two-sided Mann–Whitney
test between two diets, with the AL diet as the reference group. The solid black line
and white dot represent the interquartile range and the median, respectively.

## Calorie restriction increases adaptation to stress

Mice in this study experienced stress due to handling during an annual
1-month long phenotyping event[8] (Supplementary Fig. 1A). Handling
involved interactions with lab technicians and instruments, changes
between group and single housing, periodic availability of running
wheels, and other stressors which resulted in the disruption of steady
state homeostasis (Fig. 3E, top panel). We defined adaptation to stress
as the rate at which a mouse returns to steady state dynamics following
a deviation from steady state caused by one or more phenotyping
events. We identified time windows immediately following a

phenotyping event (or group of events) where a mouse is returning to
steady state (shaded portions on bottom panel of Fig. 3E), and mea-
sured the rate of adaptation to these stressor events. Specifically, we
modeled the steady state posterior probability following a phenotyp-
ing event using a cumulative exponential distribution function and
selected the parameter that generated the best fit to this function as
the rate of return to steady state (Methods). The higher the adapt-
ability to stress, the faster a mouse returns to steady state.

We measured adaptation to stress following seven of the ten
phenotyping assays which substantially disrupted mice from steady

**A  Graphical illustration of time-varying Cox proportional hazard model**

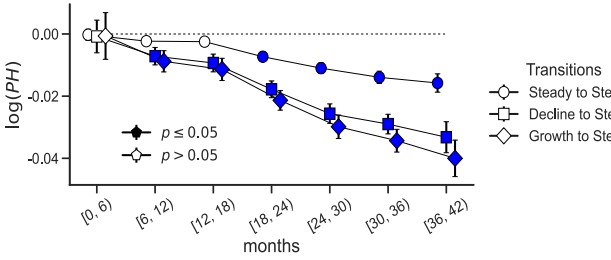

**B  Lifespan association of state occupancy**

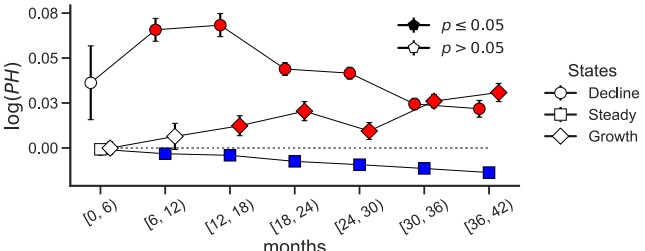

**C  Lifespan association of transitioning to steady state**

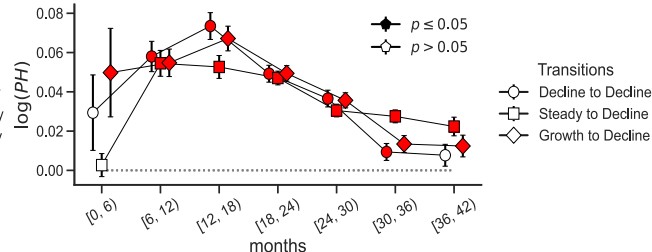

**D  Lifespan association of transitioning to decline state**

**Fig. 4 | Influence of body weight homeostasis on lifespan. A** A directed acyclic graph capturing the dependencies between diet, body weight, derived phenotypes, and lifespan. We used a time-varying Cox proportional hazard model to determine the association between body weight-derived traits and lifespan, accounting for confounding factors such as body weight and diet. **B** Effect size (+/− s.e.) of state occupancy on lifespan as a function of age. **C** Effect size (+/− s.e.) of transitions to steady state on lifespan as a function of age. **D** Effect size (+/− s.e.) of transitions to decline state on lifespan as a function of age. In (**B**), (**C**), and (**D**), markers indicate the estimated effect size, and error bars indicate the standard error. Solid markers indicate associations with $p < 0.05$, determined using the two-sided Wald test. Positive effect sizes are denoted by red markers while negative effect sizes are denoted by blue markers, for ease of visualization.

state (Supplementary Fig. 5A). Across all assays, on average, 40% CR mice showed the fastest adaptation to stress (Fig. 3F). To quantify age-related effects on adaptation to stress, we divided the age-axis into two non-overlapping bins: (6–18) months and (18–30) months; the age-bins were picked to ensure that the number of assay events in each age-bin were equal. Across most assays, we noticed that AL mice recovered to steady state more rapidly in late life than in mid life (Supplementary Fig. 5B–C), while mice on other diets did not show a significant difference in stress adaptation between mid-life and late-life. This suggests that a short duration (3–4 months) of dietary restriction induced and maintained an equivalent amount of adaptability to stress as that induced by the stressors associated with a battery of phenotyping assays spread over a year.

**Steady state homeostasis is associated with reduced mortality throughout life**

Body weight in mice is known to be predictive of lifespan in an age-dependent manner[17]; however, it is unknown whether the dynamics of body weight at different ages are also predictive of lifespan. We estimated the age-dependent effect of our derived homeostatic traits on mortality hazard (Fig. 4A), after controlling for the effects of other lifespan determinants such as diet, body weight, interaction between body weight and diet, and generation. Specifically, we applied a time-varying Cox proportional hazard model[25] to describe the survival time as a function of time-independent covariates such as diet and generation, and time-dependent covariates such as body weight (densely measured) and age-binned measures of homeostasis (sparsely quantified) (Methods).

We observed that steady state homeostasis was positively associated with lifespan and the strength of association gradually increased with age. Conversely, decline state homeostasis and growth state homeostasis were both negatively associated with lifespan, with a much stronger effect for decline state homeostasis (Fig. 4B, Supplementary Fig. 6A). The association of decline state homeostasis with

lifespan was strongest during middle age (between 6–18 months), highlighting that sustained loss of body weight during mid life has severe detrimental effects on mouse lifespan. To provide a more intuitive interpretation of the effect size, we can translate the log proportional hazard (log(PH)) into a concrete percentage change in mortality risk. For instance, during the 6 to 12-month age period, the log(PH) for the decline state is approximately + 0.065. Since state occupancy is measured as a percentage, this indicates that for each one percentage point increase in the time a mouse spent in a state of weight loss, its instantaneous risk of death (hazard) increased by 6.7% ($\exp(0.065) \approx 1.0671$). Across all three phenotypes, the effect size of 40% CR diet on lifespan was the largest. The effect sizes of body weight on lifespan were comparable in magnitude to the effect sizes of the phenotypes on lifespan. The effect sizes of the interaction between diet and body weight were significant only in the CR diets during late life (Supplementary Fig. 6B–D).

Since increased steady state homeostasis was associated with increased lifespan, we sought to quantify the effect of transitions to steady state (from other states) on lifespan. We observed that the rate of transition to steady state was positively associated with lifespan as well, although the effect sizes were dependent on the state from which the mouse transitioned (Fig. 4C). Similar to our earlier observation, high rates of transition to decline state were associated with a decreased lifespan (Fig. 4D) whereas transition to growth state was not associated with lifespan (Supplementary Fig. 7A). In contrast to our homeostatic traits, adaptation to stress (Fig. 3F) due to the phenotyping events showed no significant association with lifespan (Supplementary Fig. 7B).

**Genetic determinants of body weight dynamics are distinct from those of body weight**

Leveraging the genetic diversity in our mouse cohort, we sought to map genetic variants associated with the dynamics of body weight

trajectories. For each mouse, we split the body weight trace into a pre- and post-intervention phase and computed 19 traits for each phase. We computed the partial correlation between all pairs of these 38 traits, controlling for diet and cohort effects, and their genetic correlation using a matrix-variate linear mixed model[26]. We identified two major clusters of traits based on their genetic correlation structure. (Supplementary Fig. 8).

We carried out additive genetic mapping of these 38 traits using the GxEMM model[20]. Accounting for the correlation structure among these traits, we computed a genome-wide significance threshold of $6.30 \times 10^{-5}$ using a cluster-specific false discovery rate (FDR) control of $\alpha = 0.05$ (Methods). We applied this threshold to the post-intervention traits (Supplementary Fig. 7C), and identified 10 traits with at least one variant crossing the genome-wide significance threshold. All variants significantly associated with these 10 traits could be grouped into 8 genetic loci (Fig. 5A). None of these loci that mapped to our derived traits overlapped with genetic loci previously mapped to body weight of mice from the same experiment[20]. Furthermore, we tested for additive genetic effects on one trait in the pre-intervention phase that was associated with lifespan (maximum absolute rate of decline; Supplementary Fig. 7C) but identified no significant loci. Finally, we also tested for genotype × diet interaction effects on each of these traits but identified no significant interaction effects.

We evaluated whether the genome-wide genetic contribution to these 10 traits differed between the different diets. We quantified the total and diet-dependent heritability for each of these traits using the GxEMM model, controlling for diet and generation effects. We observed that the genetic contribution to these traits differed between dietary contexts, with traits typically being more heritable in the 40% CR diet. The higher heritability of these traits in the 40% CR diet can be largely attributed to increased genetic variation relative to other diet groups, suggesting that the dynamics of body weight are tightly regulated under severe calorie restriction. Additionally, we observed that the heritability of the derived traits were comparable to the heritability of post-intervention body weight (genetics explained 20–30% of the variation in body weight)[20] and the heritability of lifespan (genetics explained 23.6% of the variation in lifespan)[8].

To interpret these genetic associations, we fine-mapped each of the 10 significant trait-locus pairs using allele-dosage at both genotyped and imputed variants within a 1-megabase (Mb) window centered at each locus. Our fine-mapping analysis confirmed 8 significant trait-locus pairs (Table 1). We highlight the fine-mapped variants for two traits: (a) maximum absolute rate of decline, which was associated with decreasing lifespan, and (b) steady state homeostasis, which was associated with increasing lifespan (Fig. 5C–D). At the chromosome 17 locus associated with maximum absolute rate of decline, we grouped fine-mapped variants based on their founder-allele-pattern (FAP) and ranked groups based on the largest LOD score among its constituent variants, to identify variants and haplotypes most likely responsible for the association at this locus (Methods). The top FAP group contained variants with minor alleles specific to the WSB strain (Fig. 5C, Supplementary Fig. 9A). All significant variants were located in a genomic interval containing three genes: *Fer*, *Pja2*, and *Man2a1* (Fig. 5C). Intersecting the WSB-specific significant variants with regulatory elements assayed across a number of mouse tissues[27,28], we observed that the strongest variants ($p < 10^{-5}$) were located within regulatory elements around *Fer* and *Man2a1* genes, active across adipose, intestine, and muscle tissues (Supplementary Fig. 9B). These observations suggest that this locus likely influenced the rate of loss of body weight by regulating gene expression in tissues directly related to metabolism. Similarly, at the chromosome 6 locus associated with steady state homeostasis, the top FAP group contained variants with minor alleles shared by the NOD, CAST, PWK, and WSB strains (Fig. 5D, Supplementary Fig. 9C). All significant variants were located in a genomic interval containing the following genes: *Pzp* and *Nkrp1-Clr* cluster

(Fig. 5D). The strongest candidate variants were located within regulatory elements at the *Pzp* and *Clec2d* genes, active across most tissues (Supplementary Fig. 9D).

## Discussion

In this study, we derived novel phenotypes describing the dynamics of body weight measurements taken at high temporal resolution throughout life, highlighted changes in these phenotypes due to diet and age, evaluated their utility for predicting lifespan, and identified their genetic determinants. To quantify temporal dynamics in body weight, we developed an autoregressive hidden Markov state space model to extract states representing growth, decline, and maintenance of steady body weight (Fig. 2). Previous studies investigated age-related trends in body weight from longitudinal measurements using simple parametric models. Due to the non-monotonic nature of body weight dynamics, studies have fit an asymmetrical inverted-U pattern either by piece-wise linear regression or polynomial regression and dividing the age-axis into early (growth), mid (steady), and late (decline) life based on the estimated slopes[29]. Such approaches are well suited to capturing long-term trends in mean body weight, but do not capture short-term, transient changes in body weight within each phase such as loss or gain and subsequent stabilization of body weight in response to stressful life events. In contrast, the ARHMM's ability to incorporate autoregressive components directly into the hidden state dynamics allows for capturing the temporal dependencies in body weight data, while the hidden states provide a framework for identifying distinct phases or regimes in body weight trajectories. This aligns with our goal of uncovering underlying patterns and shifts in body weight dynamics that may not be readily apparent from autoregressive integrated moving average (ARIMA) and mixed-effects models. Our model of body weight trajectories as a sequence of states allowed for the detection of multiple, transient switches between these states that occur throughout life and captured the continuous nature of body weight fluctuations (i.e., did not require fixed breakpoints of the age axis). Using these inferred states, we derived novel traits that quantify homeostasis of body weight dynamics and adaptation to stress (Fig. 3).

Prior research has demonstrated that body weight during early life predicts lifespan[30,31]. Similarly, it is well-recognized that diet has an effect on body weight[14,15,32,33]. However, no previous studies have quantified the effect of diet on body weight trajectories throughout life and how features derived from such trajectories are associated with lifespan. In our work, we showed that both CR and IF increased the proportion of time spent without dramatic weight loss or gain (steady state). This was not solely due to an inability to gain body weight due to caloric constraints or a physiological lower limit on body weight preventing weight loss. We observed that mice under 1D IF were consuming the same number of calories as ad libitum mice and that mice on CR and IF did spend brief periods of time in both growth and decline states but returned more quickly to steady state compared to ad lib mice. We also demonstrated that time spent in steady state was strongly associated with lifespan in an age-dependent manner, even after conditioning on the effects of diet and body weight (Fig. 4). Deviations from steady state, captured by the time spent in either growth or decline states, were influenced by diet and phenotyping events, and also associated with lifespan.

One interpretation of deviation from steady state is a perturbation to body weight due to extrinsic stress such as a phenotyping event. This is supported by our observation that deviations often coincided with weeks where phenotyping was performed. We hypothesized that the rate of return to steady state is a measure of adaptation to stress. We note that this measure of adaptation to stress is different from resilience, a phenotype traditionally used in the field of aging. In biology, resilience is often defined as the ability to recover following an acute stress, whereas our measure of adaptation to stress does not require recovery of the original body weight, but rather a return to

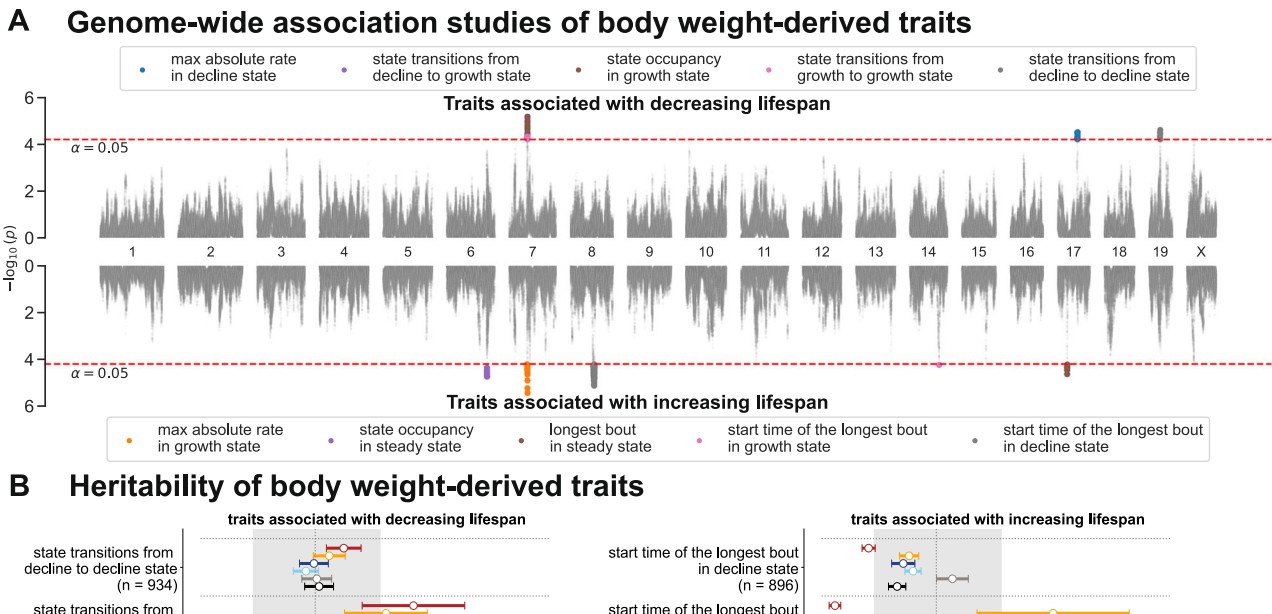

## A Genome-wide association studies of body weight-derived traits

## B Heritability of body weight-derived traits

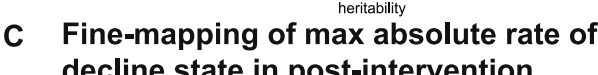

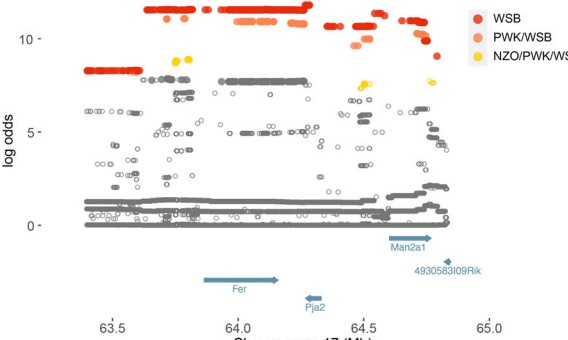

## C Fine-mapping of max absolute rate of decline state in post-intervention

## D Fine-mapping of state occupancy in steady state in post-intervention

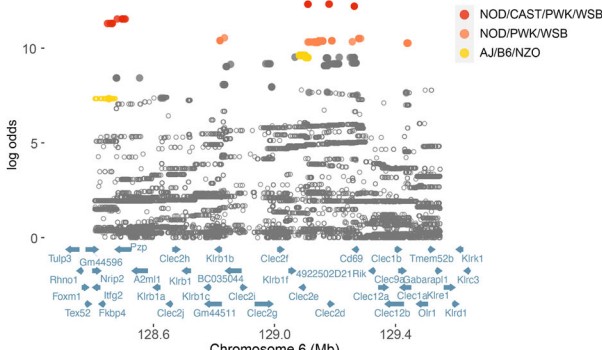

**Fig. 5 | Influence of genetics on body weight-derived traits. A** Manhattan plot of genetic associations with body weight-derived traits; traits were restricted to measurements taken after the start of dietary intervention. The red dashed line represents the genome-wide significance threshold at a false discovery rate of $\alpha = 0.05$. The $p$-values were computed using an adaptive sequential permutation scheme, permuting the phenotype vector to estimate the null distribution of the likelihood ratio statistic. **B** Forest plot of the diet-specific and total (TO) heritability (+/− s.e.) of body-weight derived traits. (left panel): traits associated with decreasing lifespan; (right panel) traits associated with increasing lifespan. Circles represent the esimtated total or diet-specific heritability and error bars indicate the standard error. The dashed vertical line denotes the heritability of lifespan and the shaded region denotes +/− s.e. of heritability of lifespan. **C**, **D** Fine-mapping loci associated with maximum absolute rate of decline on chromosome 17 and loci associated with steady state homeostasis on chromosome 6. Solid circles indicate significantly associated variants. Colors denote variant sets with shared founder allele patterns (FAP). Variant sets with the strongest evidence (FAP ranks 1, 2, and 3) are colored red, orange, and yellow, respectively.

homeostasis at a new and potentially different body weight[34]. Our definition of adaptation to stress is more aligned with the definition of resilience in the field of systems dynamics, where the resilience of a system is the rate at which the system converges to an equilibrium state (steady state) after a disturbance[35]. Thus, when a mouse returns to steady state following a phenotyping event, it is not required that the mouse regains the body weight lost during the phenotyping event.

Our findings on the unique metabolic and lifespan characteristics of female mice can be contextualized within the evolutionary

framework of the disposable soma theory of aging[36]. This theory posits an inescapable trade-off between an organism's investment in reproduction versus the maintenance and repair of its own body. The female mouse, in particular, has an evolutionary history shaped by the metabolic demands of reproduction, requiring a physiology optimized for high energy turnover[37]. Our results align with the consequences of such a trade-off. We can interpret the steady state, which is strongly associated with increased lifespan, as a period of successful somatic maintenance. Conversely, the decline state, which predicts mortality,

**Table 1 | Fine-mapping of body weight-derived phenotypes measured after dietary intervention**

| Phenotype (Lifespan Association) | Chr | FAP | FAP Rank | P-value | FAP Position | | Candidate Genes |
|---|---|---|---|---|---|---|---|
| | | | | | Start | End | |
| *Decreasing lifespan* | | | | | | | |
| maximum absolute rate of decline (↓) | 17 | WSB | 1 | 1.04E-06 | 63.863049 | 64.75511 | *Fer, Pja2, Man2a1* |
| state transitions from decline to growth (↓) | 7 | AJ, 129, NZO, CAST | 2 | 4.27E-06 | 74.217763 | 74.867976 | *Slco3a1* |
| state occupancy in the growth state (↓) | 7 | AJ, 129, NOD | 1 | 1.78E-07 | 73.602418 | 75.116272 | *Fam174b, St8sia2* |
| state transitions from growth to growth (↓) | 7 | AJ, 129, NZO, CAST | 1 | 4.29E-06 | 74.23546 | 74.897911 | *Slco3a1* |
| *Increasing lifespan* | | | | | | | |
| state occupancy in the steady state (↑) | 6 | NOD, CAST. PWK, WSB | 1 | 7.74E-07 | 128.483567 | 129.186535 | *Pzp, Clec2d, Cd69* |
| maximum absolute rate of growth (↑) | 7 | AJ, 129, NOD | 1 | 2.30E-06 | 73.970385 | 75.014986 | *St8sia2* |
| longest bout in steady state (↑) | 17 | AJ, 129, NOD, NZO | 2 | 8.03E-06 | 31.944769 | 32.389439 | *Pdxk-ps* |
| start time of the longest bout in decline (↑) | 8 | NOD, PWK, WSB | 1 | 2.61E-06 | 79.47442 | 84.496365 | *Tecr, Brme1, Cacna1a* |

represents the failure of this maintenance as the soma becomes disposable. The observation that caloric restriction dramatically increases time spent in the protective steady state[38] further supports this framework; CR acts as a signal to defer costly reproduction and reallocate resources toward survival and somatic repair. Although the females in our study were virgins, their biology is a product of these deep evolutionary pressures, meaning the trade-offs described by the disposable soma theory are intrinsically active in shaping their physiology and aging trajectory. This may provide an ultimate explanation for why the dynamics of body weight homeostasis are so tightly linked to longevity in female mice.

While most studies used inbred strains to evaluate the metabolic effects of CR and IF, in this study, we used mice from a genetically diverse outbred stock which enables us to estimate genetic effects on body weight dynamics. Out of 17 lifespan-associated traits, genetic mapping identified 10 traits mapped to 8 genomic loci (Fig. 5). One trait, maximum rate of decline in body weight, was negatively associated with lifespan and fine-mapped to the gene *Man2a1*, with the strongest candidate variants located within regulatory elements that were active in adipose, intestine, and muscle tissues. Mannosidase alpha class 2A member 1 (*Man2a1*) encodes a glycosyl hydrolase that has previously been associated with the pathogenesis of inflammatory bowel disease[39], and its inhibition ameliorated rapid body weight decline and symptoms of ulcerative colitis in mouse models. Another trait, steady state homeostasis or the proportion of time spent in steady state, was positively associated with lifespan and fine-mapped to the gene *Pzp*, where the fine-mapped variants colocalized within a regulatory element active across several tissues relevant to metabolism including adipose and liver tissues. Pregnancy zone protein (*Pzp*), a member of the alpha-2 globulin family of proteins, was recently identified as a key hepatokine regulating factor for fasting-refeeding triggered energy homeostasis through inter-organ cross talk between liver and brown adipose tissue[40]. Finally, we identified a pleiotropic locus on chromosome 7 linked to growth state homeostasis, which was associated with decreasing lifespan, and the maximum rate of body weight gain, which was associated with increasing lifespan. The association at this locus was fine-mapped to a single gene, *St8sia2* (ST8 alpha-N-acetyl-neuraminide alpha-2,8-sialyltransferase 2). *St8sia2* regulates the linkage of polysialic acid to neural cell adhesion molecule[41], and its linkage in the

hypothalamus plays an important role in several aspects of energy balance including food intake[42].

One key limitation of this study is the lack of high resolution measurements of food consumption and energy expenditure in individual mice. We used an additional 160 DO mice, subjected to the same dietary interventions at the same age, to evaluate daily and cumulative food intake[8]. We observed that mice on IF diets lost body weight during the fasting period, displayed compensatory feeding after the fasting period, and recovered their body weight. In contrast, the CR mice had their food intake carefully regulated, by design. However, on weekends, these mice consumed their triple allotment of food by Saturday 15:00 (for 40% CR mice) and by Sunday 15:00 (for 20% CR mice). Despite this period of fasting, the body weight of CR mice showed little fluctuation over the weekend. Although there was high concordance between average food consumption and our study design at young ages, individual feeding behavior can change with age, and it is possible that some of the variability observed was due to varying degrees of individual caloric intake and expenditure.

In addition to the limitations regarding food consumption and energy expenditure, body composition and adiposity are also strongly influenced by diet and energy expenditure, and are predictive of lifespan. Ideally, future studies would also include dense longitudinal measurements of body composition, food intake, and energy expenditure. Given the importance of body weight changes in response to extrinsic events, a study design that samples body weight more densely around such events (e.g., daily or more often) would allow for a more accurate characterization of adaptation to stress. A measure of how well body weight is regulated in response to stressful events could help identify factors that promote resilience. Additionally, combining sparse temporal sampling of molecular and cellular data (epigenetics, proteomics, metabolomics, and lipidomics) with the dynamics of densely sampled body weight could provide new insight into the mechanisms underlying homeostasis and aging.

## Methods

### Mouse housing, feeding, and body weight measurements

The DO mice were generated by breeding eight founder inbred strains to produce an outbred heterozygous population with a random assortment of genetic variation. These founder strains were A/J (RRID: 000646), C57BL/6J (RRID: 000664), 129S1/SvlmJ (RRID: 002448),

NOD/ShiLtJ (RRID: 001976), NZO/HlLtJ (RRID: 002105), CAST/EiJ (RRID: 000928), PWK/PhJ (RRID: 003715), and WSB/EiJ (RRID: 001145). In this study, 960 DO female mice, sampled at generations 22-24 and 26-28, were enrolled into the study after wean age of 3 weeks, and maintained at the Jackson Laboratory in Maine. The sample size was determined to detect a 10% change in mean lifespan between intervention groups with allowance for some loss of animals due to non-age-related events. No mice in the study were siblings and maximum genetic diversity was achieved. There were two cohorts per generation for a total of 12 cohorts and 80 animals per cohort. Enrollment occurred in successive quarterly waves starting in March 2016 and continuing through November 2017. Mice were housed in pressurized, individually ventilated cages at a density of eight animals per cage with random cage assignments. They were subject to 12 hours of continuous light and dark cycles beginning at 06:00 AM and 06:00 PM, respectively. Animals exit the study upon death. All animal procedures were reviewed and approved by the Animal Care and Use Committee at The Jackson Laboratory IACUC protocol 06005.

For mice on intermittent fasting (IF) diets, fasting was imposed every week on Wednesday at 15:00, for 24 or 48 h for the 1D and 2D groups, respectively. These mice had unlimited food access (similar to AL mice) on their non-fasting days (Fig. 1B). For mice on the calorie restriction (CR) diets, a measured amount of food was provided daily at around 15:00; 2.75 grams/mouse/day for 20% CR and 2.06 grams/mouse/day for 40% CR. For these mice, on Friday around 15:00, CR mice were given three times the amount of food until the following Monday afternoon (Fig. 1B). The amount of food provided to CR mice was determined based on a previous internal study at the Jackson Laboratory where the average amount of food consumed by female DO mice at 6 months of age was estimated to be 3.43 grams/day[8]. For the CR mice, food was weighed out for an entire cage of eight. Competition for food was minimized by placing food pellets directly into the bottom of the cage, allowing individual mice to 'grab' a pellet and isolate while they eat. Observation of the animals indicated that the distribution of food consumed was roughly equal among all mice in a cage. As the number of mice in each CR cage decreased over time, the amount of food given to each cage was adjusted to reflect the number of mice in that cage.

The body weight measurements for this analysis were collated in October 2021, at which point 960 mice (97.35%) had measurements at 6 months, 884 (92.08%) at 12 months, 803 (83.64%) at 18 months, 639 (66.56%) at 24 months, 416 (43.33%) at 30 months, 187 (19.47%) at 36 months, and 43 (4.47%) at 42 months. We included all body weight measures for each mouse up to 1632 days of age (>42 months). We excluded four mice from the downstream analysis (1 from AL group, 2 from 1D group, and 1 from 2D group) because these mice died early in the study and had fewer than seven body weight measurements in total, resulting in 956 mice in our analyses. To obtain a uniform sampling interval for the body weight dynamics, we rounded the time (in days) of measurement of the body weight to the nearest ten. When two body weight measurements were available at the nearest ten, we took the average of these measurements and assigned it to the nearest ten. This strategy of creating a uniform sampling of 10 day sampling interval resulted in 4.83% missing values at specific time points. Because the number of missing values is a tiny fraction of the entire database, we used a one-dimensional linear interpolation filter to fill the missing values.

### Autoregressive hidden Markov model

Autoregressive models are based on the idea that the current value of the time-series, $y_t$ measured at time $t$, can be explained as a function of $\ell$ past values, where $\ell$ is the lag-order and determines the number of steps into the past needed to forecast the current values. The simplest form of an autoregressive model can be represented as AR($\ell$) which takes the form $y_t = \phi_0 + \phi_1 y_{t-1} + \ldots + \phi_\ell y_{t-\ell} + \epsilon_t$, where $(\phi_0, \ldots, \phi_\ell)$ are the autoregressive coefficients and $\epsilon_t$ is the additive white Gaussian noise distributed as $\mathcal{N}(0, \sigma^2)$ (see Fig. 2A). A hidden Markov model is a state-space model which is characterized by the hidden states and observations generated by the hidden states. The hidden states are assumed to follow a first-order Markov chain and can only be detected through the observed sequence as they emit observations on varying probabilities. A Gaussian hidden Markov model where $s_t \in \{1, \ldots, K\}$ represents latent state at time $t$ is specified with an initial probability distribution $\pi_0 \in \mathbb{R}^{K \times 1}$, a transition matrix $A \in \mathbb{R}^{K \times K}$ with each $a_{i,j}$ representing the probability of moving from state $s_{t-1} = i$ to state $s_t = j$ such that the row sum equals one, and a sequence of observation likelihoods, also called as emission probabilities, each expressing the probability of an observation $y_t$ being generated from state $s_t = i$ drawn from a Gaussian distribution $\mathcal{N}(\eta_{s_t}, \sigma_{s_t}^2)$ (see Fig. 2A).

The ARHMM combines an autoregressive model and a hidden Markov model. In an ARHMM, the observations are generated by a few autoregressive time-series models of fixed lag-order, where the switching between these models is controlled by the hidden states which follow a first-order Markov chain. We denote the ARHMM of $l$-th lag-order as ARHMM($\ell$) and is defined as $y_t = \phi_0^{s_t} + \phi_1^{s_t} y_{t-1} + \ldots + \phi_\ell^{s_t} y_{t-\ell} + \epsilon_t^{s_t}$, where $s_t$ represents one of the $K$ possible latent states, $(\phi_0^{s_t}, \ldots, \phi_\ell^{s_t})$ are the autoregressive coefficients corresponding to state $s_t$, and $\epsilon_t^{s_t}$ is the state-dependent additive white Gaussian noise distributed as $\mathcal{N}(0, \sigma_{s_t}^2)$. To capture the variation within each latent state, we assumed that the autoregressive coefficients are drawn from a Gaussian distribution of unknown mean and variance given as $\phi_{s_t}^\ell \sim \mathcal{N}(\eta_{\ell, s_t}, \sigma_{\ell, s_t}^2)$, where $\ell$ is the lag-order of the autoregressive model in state $s_t$ (see Fig. 2A).

The physiological states are represented by the discrete latent states that capture the underlying body weight dynamics at an organismal level. Furthermore, we only considered autoregressive models of lag-order one and set the autoregressive coefficient of the first lag-order to be one, i.e., $\phi_1^{s_t} = 1, \forall 1 \le s_t \le K$. Therefore, $y_t = \phi_0^{s_t} + y_{t-1} + \epsilon_t^{s_t}$, which implies that the body weight at time $t$ depends on body weight at time $t-1$, a state-dependent random variable $\phi_0^{s_t}$ which is drawn from $\mathcal{N}(\eta_{0, s_t}, \sigma_{0, s_t}^2)$, and a state-dependent error which is drawn from $\mathcal{N}(0, \sigma_{s_t}^2)$. These simplifications made the latent states extracted using the ARHMM interpretable because the physiological states are determined based on the sign and magnitude of the mean of the $\phi_0^{s_t}$ autoregressive coefficient. We implemented the ARHMM using a varaitional inference approach. The details of the derivation are provided in the Supplementary Information.

### Adaptation to stress

We used the posterior probability of steady state as a proxy to measure the rate of recovery because mice spend a significant proportion of their lifespan in the steady state. To compute the rate of recovery, first we identified regions of homeostasis. A region of homeostasis starts when the posterior probability of steady state goes below 0.05, and ends when the probability of steady state stops monotonically increasing or is monotonically increasing and goes beyond 0.95. Thereafter, for each region, we fit a cumulative distribution function of an exponential distribution, which is given as $(1 - e^{-\lambda x})$, where $\lambda$ is the rate parameter of an exponential distribution. Finally, for any given region of homeostasis longer than thirty days duration, we had at least 3 measurements to estimate $\lambda$ and the value of $\lambda$ that generated the best fit was considered as the rate of recovery.

### Time-varying Cox proportional hazard model

In a traditional Cox model, the hazard ratio is dependent on the covariates and independent of time. In contrast, in a time-varying Cox proportional model, the hazard ratio is dependent on time, i.e, the covariates vary with time. The time-varying covariates are classified into two types: internal (dependent on individuals in the study) and external (independent of the individuals in the study). In our proposed

model for lifespan analysis, we used body weight as an internal time-varying covariate and the body weight-derived trait measured in a given time-interval bin as an external time-varying covariate. We considered the following time-varying Cox proportional model:

$$h(t, \mathbf{Z}) = \underbrace{h_0(t)}_{\text{baseline hazard}} \exp\left\{ \underbrace{\sum_{d \in D} \beta_d^{(D)} Z^{(D)}}_{\text{diet}} + \underbrace{\sum_{g \in G} \beta_g^{(G)} Z^{(G)}}_{\text{generation}} + \underbrace{\sum_{j=1}^{N} \beta_j^{(T)} Z^{(T)}(B_j)}_{\text{trait:age bin}} + \underbrace{\sum_{j=1}^{N} \beta_j^{(B)} \sum_{t \in B_j} Z^{(B)}(t)}_{\text{body weight:age bin}} \right.$$
$$\left. + \underbrace{\sum_{j=1}^{N} \sum_{d \in D} \beta_{j,d}^{(B,D)} \sum_{t \in B_j} Z^{(B,D)}(t)}_{\text{body weight : diet : age bin}} \right\},$$

where $h_0(t)$ is the baseline hazard, $N$ is the number of non-overlapping and continuous age bins represented as $\{B_1, ..., B_N\}$, and $\beta_j^{(T)}$, $\beta_j^{(B)}$, $\beta_d^{(D)}$, $\beta_{j,d}^{(B,D)}$, and $\beta_g^{(G)}$ are the effect sizes of (trait: age bin) interaction covariates, (body weight: age bin) interaction covariates, (diet: age bin) interaction covariates, (body weight: diet: age bin) interaction covariates, and generation covariates, respectively. We used `CoxTimeVaryingFitter` function from the `lifelines` package implemented in Python to estimate the effect sizes of the body weight-derived trait at every 6 month interval[43]. We set the penalty term required by this function in an empirical manner. Specifically, we partitioned our set of mice into a training and validation set (80%:20% split) and performed a grid search to select the penalty value that yielded the highest concordance index (C-index) on the validation set.

### Genotype measurements
We collected tail clippings and extracted DNA from 954 animals. Samples were genotyped using the 143,259-probe GigaMUGA array from the Illumina Infinium II platform by NeoGen Corp. We evaluated genotype quality using the R package: qtl2. We processed all raw genotype data with a corrected physical map of the GigaMUGA array probes. After filtering genetic markers for uniquely mapped probes, genotype quality, and a 20% genotype missingness threshold, our dataset contained 110,807 markers. Next, we examined the genotype quality of individual animals. We found seven pairs of animals with identical genotypes, which suggested that one of each pair was mislabeled. We identified and removed a single mislabeled animal per pair by referencing the genetic data against coat color. Next, we removed a single sample with missingness in excess of 20%. The final quality assurance analysis found that all samples exhibited high consistency between tightly linked markers: log odds ratio error scores were less than 2.0 for all samples. The final set of genetic data consisted of 946 mice. For each mouse, starting with its genotypes at the 110,807 markers and the genotypes of the 8 founder strains at the same markers, we inferred the founders-of-origin for each of the alleles at each marker using the R package: qtl2. This allowed us to test directly for association between founder-of-origin and phenotype (rather than allele dosage and phenotype, as is commonly done in QTL mapping) at all genotyped markers. Using the founder-of-origin of consecutive typed markers and the genotypes of untyped variants in the founder strains, we then imputed the genotypes of all untyped variants (34.5 million) in all 946 mice.

### Genetic mapping with genotyped markers
For each phenotype, we tested for additive effects between the founder-of-origin at each genotyped variant and the phenotype using the GxEMM model[20], controlling for diet and cohort as fixed effects and allowing for genotype-x-diet random effects. Expecting that the set of phenotypes in our genetic analyses were not all independent, we adopted a data-driven approach to determine genome-wide level significance threshold. First, we computed the phenotype-genotype correlation matrix for all derived phenotypes (Supplementary Fig. 7C).

Clustering of the phenotypes using their genetic correlations resulted in two major clusters (Supplementary Fig. 8); we controlled for false discoveries using the following procedure within each cluster.

For each variant, we controlled for false associations with phenotypes within a cluster, using $\alpha = 0.05$. For variants that were not significantly associated with any phenotype, we assigned a $p$-value by randomly sampling from its $p$-values for *all* phenotypes within the cluster. For variants that were significantly associated with at least one phenotype, we assigned a $p$-value by randomly sampling from the *subset* of $p$-values for significantly associated phenotypes in the cluster. This procedure generated a cluster-specific $p$-value vector across all variants. Finally, using this $p$-value vector, we controlled for false discoveries genome-wide, at $\alpha = 0.05$, to obtain a cluster-specific genome-wide significance threshold. The minimum significance threshold across both clusters was used as the genome-wide significance threshold for the study. This procedure resulted in a conservative threshold of $6.30 \times 10^{-5}$ (at $\alpha = 0.05$).

### Genetic fine-mapping with founder-allele-patterns
To more precisely fine-map the genomic interval of each QTL, we imputed all SNPs and insertion-deletion variants from the fully sequenced DO founders for a 5 Mb interval centered at the lead genotype-marker. For each imputed variant, we identified the founder-of-origin for the major and minor allele. To illustrate the process, consider the bi-allelic A/G variant. If allele A was specific to founders AJ, NZO, and PWK, and allele G was specific to the other five founders, then we assigned A to be the minor allele and defined a founder-of-origin allele pattern (FAP) of AJ/NZO/PWK for this variant. We identified variants and founder haplotypes most likely responsible for the association at each locus by grouping variants based on their FAP and ranked groups based on the largest logarithm of odds (LOD) score among its constituent variants. We hypothesized that the functional variant(s) responsible for trait-specific variation were among those in the lead FAP group because they exhibit the strongest statistical association and it is unlikely any additional variants are segregating in this genomic interval beyond those identified in the full genome sequences of the eight founder strains. By focusing on the FAP groups with the largest LOD scores, we significantly reduced the number of putative causal variants, while representing the age- and diet-dependent effects of these loci in terms of the effects of its top FAP groups. We further narrowed the number of candidates by intersecting the variants in top FAP groups with functional annotations (e.g., gene annotations, regulatory elements, tissue-specific regulatory activity, etc.).

### Reporting summary
Further information on research design is available in the Nature Portfolio Reporting Summary linked to this article.

## Data availability
The phenotype data, which includes both raw body weight measurements and calculated body weight-derived traits, can be accessed publicly at https://github.com/calico/do_bwd/tree/main/data. Genotype data was downloaded from our previously published DRiDO study[8] and is available for download at https://doi.org/10.6084/m9.figshare.24600255.

## Code availability
The code for extracting body weight-derived traits using an autoregressive hidden Markov model, as well as the subsequent lifespan and genetic analyses of these traits, is publicly available through the following GitHub repository: https://github.com/calico/do_bwd. The repository also contains trained models, processed datasets, and scripts that can be utilized to train models from scratch and generate

processed datasets. All source data that are necessary to interpret, verify and extend the research in this article are included in these links.

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

## Acknowledgements

We thank the JAX Nathan Shock Center Animal and Phenotyping core team for their assistance with animal husbandry, data collection, and curation. We would also like to thank Eugene Melamud and Jeremy Willsey for helpful feedback and discussion. This work was funded by Calico Life Sciences LLC.

## Author contributions

G.V.P. was responsible for conceptualization, formal analysis, methodology, software, visualization, writing—original draft, and writing—review and editing; Z.C. contributed to formal analysis, methodology, supervision, writing - original draft, and writing—review and editing; K.W.

performed formal analysis, software, visualization, and writing - original draft; A.D.F. performed formal analysis, writing— original draft, and writing—review and editing; V.J. handled formal analysis, methodology, and supervision; G.A.C. contributed to formal analysis, writing—original draft, and writing—review and editing; and A.R. provided conceptualization, formal analysis, methodology, software, visualization, and supervision, in addition to contributing to the writing—original draft and writing—review and editing. All authors read and approved the final version of the manuscript.

## Competing interests

G.V.P., Z.C., A.D.F., and A.R. are employees of Calico Life Sciences. The other authors declare no competing interests.
