## [Transparent Peer Review file · Nature Communications]

Longitudinal analysis of body weight reveals homeostatic and adaptive traits linked to lifespan in diversity outbred female mice

Corresponding Author: Dr Anil Raj

Version 0:

Reviewer comments:

Reviewer #1

(Remarks to the Author)

In their manuscript, Prateek et al study the interaction between genetics and diet in determining lifespan. They split nearly one thousand outbred mice across 5 dietary regimes including various degrees of intermittent fasting and caloric restriction, and make daily measurements of lifespan from six months until 42 months of age. This is impressive data, both in its scope and in the quantitative quality of the data collected. The authors genotyped all animals using a microarray, which in combination with fully-sequenced inbred founders of the outbred population, allowed fine-grained evaluation of phenotype-genotype correlations.

The authors discretize their body-weight trajectories using a hidden Markov model that seems well suited for the task. They find, not unsurprisingly, that the degree of calorie restriction correlates with the time spent in various body weight states—gaining weight, stable weight, and losing weight. The authors then find that some aspects of body-weight dynamics during aging are associated with longer life, and identify genes associated with such body weight dynamics.

The analysis appears rigorous but throughout the manuscript, the authors appear to deliberately avoid the most important question: what has this study taught us about the relationship between diet and lifespan? Remarkably, there is no heritability analysis of lifespan which naively one would suspect to be the main question being addressed by a study of this design.

In many places, the authors seem to be actively obscuring the true effect sizes of the effects of various diets on body weight dynamics and lifespan (see points 1 and 2 below), and in general, the authors can do a much better job clarifying for the reader the magnitude and quantitative importance of their findings.

Include other effect terms in fig. 4.: e.g. the direct effects of diet and body weight on lifespan

PH ratios are useful, but What fraction of lifespan variance is explained by the heritable factors driving the dynamics of body weight? Fig 5b should show this, but is heritability in % or a fraction? How does this compare to the total heritability of lifespan across all loci (irrespective of their mediation via body weight dynamics)? Is the total heritability seen in this experiment similar to other estimates of heritability of lifespan in mice?

In the absence of a clear statement of the overall magnitude of the effect sizes being studied, the reviewer is left uncertain about how much concern should be placed on potential confounding factors. Across such a large cohort of animals, could confounding environmental factors be driving some of the associations identified?

Major points

1. The authors calculate statistics on body weight trajectories, for example, the longest continuous bout and the maximum rate of change of body weight. It's unclear how interpretable the statistics based on these extreme outliers are—for example, reasoning based on changes in “the average longest stretch of maintaining body weight” seems rather baroque. Are their simpler statistics, less likely to depend on the values of extreme outlier events, that can demonstrate the author's points? The reader is left wondering whether these exotic statistics were chosen to hide the fact that more straightforward statistics would leave the reader underwhelmed at the magnitude of the biological effects measured. Though the authors provide p-values for the statistical significance of these effects (in Fig. S3), it would probably be more helpful to readers to know the error in the estimate of the effect sizes, by providing confidence intervals.

2. In Figure 4, the authors only provide the estimated hazard ratios for one arrow of their full model—the estimated direct effect of phenotype on lifespan. This leaves the reader unclear on the magnitude of these effects relative to other effects in the model, in particular the hazard ratios for diet's effect on lifespan and body weight's effect on lifespan. Can the authors include these direct effects somewhere prominently, to provide context for the phenotype->lifespan effects?

3. The presentation of the heritability of these traits is somewhat unclear to a non-expert reader. For example, this reviewer would have expected this paper would address the heritability of lifespan itself, which presumably will be low in outbred mice just as it is in human populations. Instead, the authors discuss the "heritability of traits associated with decreasing [or increasing] lifespan" which seems a bit of a confidence trick, talking about heritability and lifespan in the same figure heading but then presenting no data at all that quantifies the heritability of lifespan. This reviewer finds an analysis of the heritability of lifespan (which, for example, might be altered by the environment, with starvation increasing the influence of some SNPs on lifespan or decreasing the influence of others on lifespan) striking in its absence. Discussing the "heritability of traits" that are themselves "associated" with lifespan seems to be substituting a murky, indirect association in place of a much clearer analysis of the heritability of lifespan itself across the different conditions.

4. Following up on the previous comment, the authors should make it clear what the overall heritability of lifespan is in this experiment. If lifespan has a very low or even 0 heritability in this experiment, this should be stated prominently.

5. The authors should include survival curves of some sort for all populations, to help readers and reviewers to estimate the quality of the underlying data. Presumably, the almost one thousand mice were not measured in a single age-synchronous cohort. Could batch-specific effects be influencing the findings and acting as a confounding factor in statistical analysis? For example, what is the effect size of the birth month on the time-dependent hazard rate?

(Remarks on code availability)

The code looks fine.

Reviewer #2

(Remarks to the Author)

This paper investigates the impact of dietary interventions, specifically calorie restriction (CR) and intermittent fasting (IF), on body weight dynamics and lifespan in genetically diverse Diversity Outbred (DO) mice. Using advanced statistical models and gene mapping techniques, the study identifies key genetic loci associated with body weight regulation and longevity. The research provides valuable insights into the genetic and environmental factors influencing aging and metabolic health.

Comments and Suggestions

1. Generational Differences:

The study involves DO mice from different generations (22-24 and 26-28), which could result in body weight variation due to genetic and environmental factors specific to each generation. Given the current mouse housing and breeding setup, mice with earlier reproduction have a higher possibility to pass their genes to the next generation, which may also be associated with altered body weight.

Suggestion: Consider including generation as a variable in the analysis to assess its impact on body weight variation. Additionally, discussing potential reasons and implications of generational differences in body weight will help clarify if and how generational effects influence the observed outcomes and ensure the robustness of the study's conclusions.

2. Food Intake Adjustment:

The method of adjusting food amounts based on previous studies where "female DO mice at 6 months of age were estimated to consume 3.43 grams/day" is not scientifically rigorous. This is because food consumption in DO mice may vary significantly due to genetic diversity and may also vary with aging.

Suggestion: First, discuss the potential food consumption variation due to genetic differences and changes with age. Second, consider renaming the treatment categories; instead of using 20% and 40% food restriction, use terms like "mild" and "severe" restriction to better reflect the nature of the intervention.

3. Alternative Mathematical Models:

Since part of the readers of this paper will be gerontologists with a background in biology, it would be beneficial to discuss other potential mathematical models that can be used for determining the trend of body weight, besides the hidden Markov model used in this study.

Suggestion: Include a brief discussion of alternative models such as autoregressive integrated moving average (ARIMA) models, state-space models, and mixed-effects models. This comparison can provide additional context and support the robustness of the chosen methodology.

Overall, the authors provided a comprehensive and rigorous analysis of the effects of CR and IF on body weight dynamics and lifespan. The use of genetically diverse DO mice enhances the generalizability of the findings. The integration of advanced statistical models and gene mapping techniques strengthens the validity of the conclusions. This study makes a significant contribution to the understanding of genetic and environmental influences on aging and metabolic health.

(Remarks on code availability)

After a quick review of the provided code, it appears to be well-documented and potentially reproducible. The code includes a README file with clear instructions for installation and running the application, which enhances its usability for the community. Although I have not personally tested the code, the documentation and structure suggest that it should be repeatable. However, a thorough validation by running the code would be necessary to confirm this assessment fully.

Reviewer #3

(Remarks to the Author)

Overview of Review. First, apologies in advance to the authors for some redundancy in these comments.

This is an excellent contribution to our understanding of the associations and causal linkages between body weight changes as a function of diet and four methods of dietary restriction. There is much gold here, but it takes a persistent reader or reviewer—one able to cycle through paper twice to recognize the merits of the work. This should be relatively straightforward problem to address.

It is important however to optimize this paper so that it will have the impact it deserves. Many detailed suggestions below.

One of the major problems is that the introduction of this MS fail to highlight 1. the design, 2. the biological objectives, or 3. the main conclusions. Instead, the introduction focuses on what are relatively generic technical and computational issues. The technical and mathematical treatment is overwrought throughout this manuscript given the nature of the data.

Ideally, the introduction should also explain that this submission is a companion to at least two other papers widely circulating (and well received) on bioRxiv, including the Francesca paper cited by the authors.

Are the raw or processed data as shown in figure 1D available in supplements? The coverage of lifespan data is somewhat cryptic/hidden in this paper. We believe that the animals used here are same as those in both Francesco and Luciano papers. If true or if note, this needs to be stated explicitly.

More important general (G) suggestions many regarding the introduction

G1. A very general comment on the submission: You want to ease readers into the biological importance and predictive utility of more complex derivatives of weight over lifespan as a function of diet and mild stressors at about 22 and 25 months. You do not want to confuse or impress readers with models that are actually very simple. Hold their hands. Guide them from simplicity to (modest) complexity. This is especially true if you want to be published in a general science journal like Nature Communications and to be cited heavily.

G2. Along these lines: the introduction is not interesting, and it does not do justice to the data and interventions. Like the start of the discussion (more below), it is much too focused on technical and computational details. What hypotheses are you testing or what are you trying to predict (lifespan mainly, or GXE on lifespan)?

In the last paragraph of the introduction the words “measured”, “developed”, “characterized”, “demonstrated”, “explored” and “examined”, are used but most readers will want more than this—they will want and expect at least one or two new concepts and insights. This paper is not just about a fancier quantitative approach to parse and map higher moments of body weight. Biology, not methods and tweaks of Cox proportional hazards models. Although, I will add if this new method gets around the “proportional” assumptions is itself worth a paper.

Here are some concrete suggestions for a potentially stronger introduction. Introduce your design, the population, the actual broader context (this is one of at least three papers) and the questions you specifically address and answer. You do not need a long wind-up on the basics of aging and homeostasis. Focus on the baseline dynamics of body weight change and on the main levels of perturbations—diet and genetics—as predictors of lifespan. Deemphasize the “high frequency” measures of body weight, which are at best only of moderate frequency. Some of us long-lived organisms weigh ourselves once or twice a day, and we will not be impressed by weighting once ever 10 days in no known phase with the cyclicity of four of the five diets. The “stress” mentioned a few times in this study is almost completely peripheral consequence of testing of the subset of animals that survived to 23 and 26 months. Be frank about this and explain the systematic weight losses at earlier ages.

G3. The introduction does not introduce the key dietary manipulations or their motivations. Nor does it introduce the testing perturbations at 22 and 26 months. These topics are worth at least a full paragraph—diet and genotypes are instrumental variables, and the interventions at 22 and 26 months are a mild stressor applied when the mice are getting old.

G4. Since you only used females, this needs to be stated up front. This crucial “detail” should be in the abstract and also in the introduction. There are several good reasons for using only females. These statements and justifications must be stated sooner rather than later.

G4b. Much of the paper talks about homeostasis of body weight as if this were a goal in life. It is not, and this is particularly true of female mice when reproducing. So not only are you studying one sex, but it would be good to acknowledge that female mice are “built to produce offspring” and this is inherently a process that is associated with large fluctuations in the weight and metabolic state. I am not really criticizing the experimental paradigm, but I am saying that the fixation on “homeostasis” may be misplaced. What may be a better rubric is “metabolic resilience” or anti-fragility, and here the ability to

maintain virgin female body weight may reflect resilience of the same type that a breeding female producing 5 litters over 8 months may need—even if her body weight oscillates at least over a range of 5 gm. Selection does not care about weight, it cares about fecundity and fitness.

G5. You have used state-of-the-art mapping methods, but it is not clear that you have identified “genetic determinants” of derivatives of body weight. The conventional alpha level for a genome-wide test is .05 and you have shifted the goal to 0.10. I can sympathize with this, but others may be less sympathetic. In the methods there is nothing on the definition of support intervals, and at these modest linkage statistics they will be comparatively large.

Therefore a bit mystified by the assertion that by fine-mapping you can get down to between 1 to 3 candidate genes. That means less than 100 Kb—beyond implausible with 954 mice divided into five very different diets with 64 haplotype combinations and with traits that are anything but Mendelian and with highly variable MAFs. Without a deep and very compelling dive into the support intervals, I would advise much more caution in claims of getting down to X to Y numbers of candidates for loci, and where there is a considerable risk that even small loci are the result of oligogenic variant “good luck” ascertainment bias. Look at what happened to Mackay’s *Drosophila* lifespan loci.

G6. And on the same general topic—the current MS does not provide maps for the primary measurements—body weight itself (let alone any details on lifespan—I assume perhaps to avoid overlap with the Francesca paper?). The reader has no basis to determine if derivative measurements are truly adding all that much. I think that I am close to being convince, but it would be incredibly helpful (to the point of being essential) to provide the map for the primary measure of weights across ages and diets—perhaps even of the easy to understand weight changes per defined age interval. Also of interest to understand the correlation of body weight to its derivatives, prior to any “clean up”.

G7. The last sentence of the introduction is problematic and includes the word “can” [and a typo]: “We expect that body weight measured at high temporal density can reveal patter[n]s of change that are associated with longevity and healthy aging.” Instead, you should ideally tell us something more positive and concrete—what are these patterns? What have you discovered?

G8. Figures 3 and 4 are the heart of the biology of this paper. In contrast, Figure 2, is a kind of geeky figure. It will be almost self-evident to most readers that the first derivative of body weight would have probably done the trick. In Figure 2B, the first example of a declared state transition to “loss” at 3.5 months, is surprising because weight is steady during this entire period and for a month before. (I realize this is just an example, but it is the first part of Figure 2B that anyone will look at closely.) The other “loss” states are correct. The question is why we have these first “loss” states occurred at all? One could suggest stochastic unhappiness of an individual mouse, but Figure 1D belies this idea since the drops in weight are seen across all diets at circa 11 and 15 months.

G9. I could see Figure 2 as being mildly useful in the supplement. Moving it out would enable room for one more figure focused on biology. Simple things, like body weight QTLs and KM functions and actual individual data across diets on the correlations between state occupancy and longevity. What would also be most interesting is the accuracy with which the mortality risk for an individual when they inevitably get close to the mortality cliff. This is evident in KM curves and that is certainly prominent in the details of Figure 1D. I realize this is part of the Feb 8 bioRxiv preprint by Luciano et al. Just ping that paper.

G10. Sorry, some redundancy in this comment. The temporal resolution of body weight data is not high enough to provide much in the way of bragging rights, especially since there is no claim to even pick up estrus cycle or diurnal effects. It boils down to three key states that anyone could pick out by eye or probably by taking $d(BW)/dt$. Do we really consider weights at 7 to 10 days high temporal resolution? Even daily weights are not high temporal resolution. Yes, I understand that this is 3 to 10 times higher than most previous work, but it turns out that it is not necessary for the main argument on stability of weight as a key predictor of lifespan. And the only true dynamics that were controlled at 22 and 25 months do not appear to play a prominent role in any of the results. However, this raises a potentially more interesting point on “homeostasis” versus anti-fragility. We metazoans need to be resilient, and a female mouse needs to be super-resilient during the first year of life when her fitness is tested most aggressively by all kinds of perturbations in the real world. This is why I disagree somewhat with the focus on homeostasis in this paper. The body weight of a wild non-virgin female mouse is anything but a regulated variable.

G11. The current MS overstates some key genetic findings, and should be a bit more forthright about findings that are significant using standard criteria (95% confidence of association) and those that are interesting but require validation (< 95% confidence). Another issue is that the reader does not have access to information about loci for body weight done across time. As a result, the reader cannot be certain that the loci found for the derived traits would not also have been detected for variation in body weight at different time points during the experiment. The paper should go into more detail on how FDRs were computed, or if the FDRs are accounting for genome-wide significance only. There is also the issue that the first moment (body weight) has been processed so as to produce 18 derived (independent) traits for mapping. In theory, but rarely in practice, there could be a correction for the multiple derived phenotypes. A hat tip to this issue might be classy addition.

G12. This really confused me when starting my review of the submission—the unexplained losses of weight losses at 11 and 15 months. This needs to be handled well in the paper. The weight losses due to stressful testing at 23 and 26 months will make good sense to the reader but they must needs to be marked on figure 1C (as done for the tests on the plots as in Figure 3E).

Results

R1 Minor. First sentence of results is either unneeded or must be documented. What is the error term of reweighing the same mouse every 10 minutes over an hour, or every hour over 24 hours? What level of accuracy and precision do you achieve? 0.5 g seems a reasonable goal.

R2. Figure 1. Lovely overview of the design. If all cohorts that are part of the 960 DO sample were entered into the study during the same month or season that might be a worthwhile detail to add.

R3. How do you operationally define a month in days. This problem is trivial at one level but is also why some prefer days or weeks which are not as fuzzy in their definition.

R4. Figure 1. Interesting that the 1-day/week fasting killed off an inordinately large number of females.

R5. Fig 1A. Does “M” stand for mouse? This is a minor confusion since you did not use an males. If you mean numbers of mice just use the conventional symbol N. Not sure we need the mouse SVG, but what we definitely do need is the addition of the testing period shown in Figure 3E.

R6. Fig 1C. Please add tick marks for every month. What happened at 11 months and 23 months in the colony to cause such consistent drops in body weight across all cohorts? This is not normal and suggest a strong uncontrolled annual environmental perturbation. I would deduce that all cohorts maintained in one room.

R7. Fig 1D. This is a critical figure and deserves much more room and contrast. The X-axis definitely needs an added “month-of-year” axis. The age-specific (or partly environment-season-specific) variance in body weights is what impresses me the most and there are clear wave patterns of changes across diets, most notable at roughly 11 and 14 months. 23 and 26 months. Were the diets initiated at precisely the same age and month? (Let’s see if this is discussed in terms of derivative measures.)

Ideally many more individual traces would be resolvable. Death points need to be highlighted in some way (dots). The figure could benefit from an overlay of a KM-type survival estimator. Most females drop weight sharply before death. As one would expect the body weight drop is much reduced in the “40” group that do not have weight to spare, although it is easy to see the excessive deaths in this group at circa 23–24 months.

R8. Fig 1D. In both the “20” and “40” cohorts a subset of females manages to become and maintain relatively obesity out to 24 months. Does this indicate that they are dominant females able to consistently obtain more food than their cage mates?

R9 Minor. P 4, line 45. The inbred founders of the DO need to be listed correctly. This nomenclature is wrong: AJ, B6, 129, NOD, NZO, CAST, PWK, and WSB. Please ensure RRIDs for founders and DO are listed in Methods.

R10. Minor. P 4, line 48. “At 6 months of age, randomly assigned to groups of mice were switched...” Delete “to”

R11, P 4, line 61. Please express “largest” quantitatively (variance explained will do). The “substantial” other variation is a confound of age, season, and external environmental factors. At this point the careful reader has examined Fig 1 and have questions about the sources of variance that are not related to age or even diet.

R12, line 63. “Physiological”? What is the AR-HMM actually capturing—seasonality, testing, personnel changes, humidity fluctuations? All of these will of course have an impact on the physiology and anatomy of mice. Why should the reader assume hidden physiological state changes that are relatively uniform across all diets except the most extreme? Consider this comment a restatement of the puzzle of the weight losses earlier in life.

R13, P 4, bottom paragraph. Many of your readers would benefit to an introduction to latent states—not as purely statistical entities but in the way a biologist will understand. You need to motivate the abstractions and explain why simply taking a derivative of body weight by time or month or estrus cycle is not as useful. If latent states just translate to “gaining”, “stable”, “declining” then say so. If not, explain. Occupancy time in state? Read Judea Pearl Book of Why if you need good examples of bringing statistical modeling concepts down to earth.

R14, Fig 2B. This figure would also benefit from tick marks per month and a subsidiary X-axis that is month of year.

R15, P 6, lines 87-92. This is wonderful text that belongs in the introduction. Autopoiesis is much more important than body weight homeostasis. In fact, body weight is a strongly modulated variable around an intentionally wide range in order to regulate metabolic function. ZJ Zhao and J Cao is just one example of work that may be relevant to some aspects of this submission (PMID:19447188). Hibernating species and many bat species take this to the extreme and also strongly modulate metabolic function daily or annually. Good data on non-commensal rodents such as *Peromyscus*. Entire books on restriction (see FH Bronson (1989), L Fishbein 1991).

R16, Fig 3E. Again, why so exceedingly stingy with x axis tick marks and labels. Two ticks for at 30+ month axes is beyond sparse.

R17, Fig 4A. This is more than a directed acyclic graph. It is an assertion of your model. Please put the indicator variables on top (diet). You also have a 100K genotypes as potential indicator variables (perhaps put them in a cloud). Body weight, body weight dynamics (not just “phenotypes”), and other as candidate mediators, and lifespan is the outcome of interest. Since you have at least three states you regard as mediators that are explicitly plotted in B, C, and D. Why not model them explicitly and why not test as an SEM? If you make these changes then Fig 4B, C, and D will make much more sense and may become a classic.

R18. Regarding figure 4A, you make an inference which is unclear to me. They state these derived phenotypes are lifespan-associated, but (early life) body weight itself is also a predictor of lifespan. The statement: "We estimated the age-dependent effect of our derived homeostatic traits on mortality hazard (Figure 4A)" conflicts however with figure 4a which shows the infographic of their assumption and model. In other words, figure 4A does not show the age-dependent effect of the derived homeostatic traits on mortality hazards, and it does not differentiate the effects of the derived traits relative to the age-dependent effect of body weight on mortality hazards.

R19, Figure 5a: Only five loci are found to be significant at the conventional $p < 0.05$ threshold, which means that there are 6 / 11 loci which are suggestively associated at $P < 0.10$. Furthermore, please make difference between the two associated traits on the X chromosome more obvious. It took several minutes to distinguish the two separate loci on X.

R10. Supplemental Fig 6: How are the derived traits associated/correlated with the body weight? Please add the correlation to body weight to the figure. This correlation between the derived phenotypes and the body weight could be discussed in the results section, since now it is unclear if body weight itself would have captured the same effects.

Methods

M1. The Methods do not actually describe the process and frequency with which mice were weighed. The most detail I can find is on page 4, line 50: "...approximately once a week". No information on time of day, handling, taring, or precision and accuracy. Since two of the diets involve intermittent fasting days, and the caloric restriction involves a sort of binge feeding over the weekend, these details really matter, if and only if one insists on talking about high frequency longitudinal weight data acquisition.

Page 16, line 301 describes a data collating process and numbers of cases at 6-month intervals. Please refocus on the basics since this work will be read independently of other studies.

M2. Six generations were sampled. There are five treatments. While there are 6 generational cohorts, the only cohorts that really matter much are the dietary treatments.

M3. There is no text in the Methods on the calculation of support intervals, but there is a most improbable claim that the candidate intervals contain merely 1 to 3 genes. That implies a support interval of about 50 to 100 Kb—a miraculous level of precision using fewer than 954 highly recombinant DO mice treated using 5 very strong environmental perturbations.

M4. On line 185 the authors promise to give more detail on their FDR correction in Methods. However, this part of the analysis discussed in the Method section, leaves more questions than it answers. To the reviewer it is unclear how it was done. e.g. "Expecting that the set of phenotypes in our genetic analyses were not all independent, we grouped the phenotypes into four groups to determine a group-specific genomewide-level significance threshold." Why four groups? How were the phenotypes grouped together? Why make a difference in which p values are selected in the group based on rejection of the null hypothesis (maximum p value, versus random p value). Furthermore, since there are four "independent" groups, why was the FDR applied to the α to the p value array of each group separately? Since we're talking about body weight derived traits, should the FDR not correct across all four groups to minimize potential false positives?

Discussion

D1. The first paragraph focuses almost exclusively on the modeling of body weight change and the relatively simple definitions of states—gaining, maintaining, and losing weight. What is far more interesting to biologists will be the impact of the diets and of the testing perturbations on body weight changes across the five diets and lifespan.

D2. This sentence in the second paragraph is genuinely tantalizing: "However, no previous studies have quantified the effect of diet on body weight trajectories throughout life and how features derived from such trajectories are associated with lifespan."

I have highlighted in italic what seems to be of greatest interest. But only one of the following sentences carries this idea forward, and without much added value: that longer occupancy in steady-state is associated with longer lifespan. Figure 4 is an indirect method of displaying this key result. There would be much more familiar ways of making this point, even something as conventional as survival curves by diet but more importantly by state occupancy. In other word, a plot with actual lifespan on the x axis to help readers "get it".

D3. The state changes are implicitly and sometimes even explicitly linked in this stressful life events. But there is to independent stressful life "event" but rather the imposition of a potentially stressful fixed change in the diet. The cyclicity of the diets is of course a strong potential cyclic stress that can be used as an independent variable. But here we expect daily weight data or at least three weightings per week to detect the cyclic body weight responses to dietary stress. Since the

animals are group housed, conspecific dominance hierarchies are a highly likely source of stressful events, but these are not measured in the current study. The three states—steady, growth, decline cannot be ascribed to any independent variable in this study, although to this reader's eye it looks like the main effect may be seasonal or vivarium related. The within-individual fluctuations in body weight are not exposed with any clarity.

Minor Comments on Discussion

D4. Minor. Given all we know about the temporal changes in body weight with age, it might help to clarify the benefits of 7–10 day intervals between weightings. Hourly over 24 h periods have clear biological significance. Your chosen intervals of 3 or 4 times per lunar month has potential biological significance (perhaps as a function of estrus). But most researchers would assume, perhaps incorrectly, that every 3 to 6 months is adequate. So the question is why you need this? I suspect most of us would just use a smoother over data at this high a resolution.

D5. Minor. Use of the word “healthspan” on p 2 line 16. There are now some good references in which “healthspan” has been operationalized in mice. The dynamic traits in this manuscript could become part of a non-static definition of healthspan. As always, there is some risk of circularity. For work on healthspan estimation in mice see the longitudinal work by D. Lamming and Ehninger.

D6. Minor. The qualifier “In general, non-invasive procedures” would apply to almost all mammals except mice and rats. Invasive and longitudinal procedures are precisely an advantage of rodent models in which genotypes can be replicated. Xie and Ehninger's study in Nature Communications is an excellent example. The hybrid cross-sectional and longitudinal approach is also a strong advantage of DO, UM-HET3, and of course any set of inbred strains.

D7. Minor: p 2, line 19-20. Good spot to add some equivalent mouse studies; Xie et al. in particular.

D8. Minor: p 2, line 23. Also cyclical by day, lunar month, years, and of course reproductive cycles. There is some work on life history and body weight fluctuations.

D9. Minor: p 2, line 25. This whole paragraph is bland to the point of being almost unreadable. Only the last sentence is crucial. Why not start with that as the topic sentence? The choice of references is amusing—citing papers from 2022 and 2014 as support for fundamental statements. A better selection for key papers and reviews on these topics would provide readers with more background on the importance of GxE. Perhaps cite SG Wright in addition to KM Wright since you are using derivatives of methods he developed. And one paper on diet (Yang et al. 2014) having an effect on body weight is an odd and arbitrary choice given the vast literature on this topic. The work of W. Atchley, Cheverud, Mackay, and a huge human literature seems more appropriate in this context.

D10. Minor: p 2, line 35. Tense change from past to present (derived?)

(Remarks on code availability)

Question for Nature Communications editors

1. What about data availability for individual animals? I see a section on code availability, but no promises or hints regarding data (yet?). This is a serious issue given the this paper is submitted by a for-profit company that may do its best to try to maintain the data as proprietary. If the entire data set on lifespans and body weight are not provide, then I personally regard this as unpublishable.

2. What editors are handling the other two papers associated with these data in bioRxiv. Ideally, the review of these papers would be coordinated as some level. There will be at least three papers, and of the three that I have now read this is probably the least polished.

Reviewer #4

(Remarks to the Author)

(Remarks on code availability)

Code quality is solid, helpful readme

Version 1:

Reviewer comments:

Reviewer #2

(Remarks to the Author)

The revised manuscript by Prateek et al. investigates the effects of dietary interventions on body weight dynamics and lifespan in genetically diverse Diversity Outbred (DO) mice. I have carefully considered the authors' responses to the

reviewers' comments, including mine.

The authors have addressed the main concerns raised during the initial review, providing useful clarifications regarding generational effects, variability in food intake, and the choice of statistical models. These additions enhance the transparency and interpretability of the work.

While a few minor points could still be refined, I believe the revised manuscript presents a more coherent and accessible account of the study. Notably, this work complements the recent Nature publication by Di Francesco et al. ("Dietary restriction impacts health and lifespan of genetically diverse mice," Nature 634, 684–692, 2024), which reports findings from the same mouse population. Together, these studies provide a broader view of how dietary interventions influence aging and metabolic outcomes in a genetically diverse context.

I'm in favor of publication and believe this paper will be intriguing to scientists in the relevant fields

(Remarks on code availability)

Reviewer #3

(Remarks to the Author)

Much improved. this MS was solid in its first draft but is now covers the complex design and goals somewhat more effectively

Abstract: First sentence a beast.

Introduction: Thanks for adding "female". But this is only needed once. What is needed is some consideration in discussion (optional as far as I am concerned) as to what your results imply about the metabolic demands on the sex that does all of the heavy lifting wrt reproduction.

"Enrolled" is not the right word for mice. Perhaps "enter" works better. But I like the idea of enrolling mice.

Greatly appreciate the addition of experimental design details at the start of the results and in Figure 1.

Results read well. My original plea was for a re-expression of higher order metrics on weight dynamics and linkage to lifespan that are parseable in terms of days per "whatever higher order unit" of weight you care to highlight. Reading this paper is a pleasure in some ways but a frustration in the sense that I do not have an intuition about effect sizes or ranges of differences in unit that most reader will be able to understand. Much hard core experimentation in this study but it reads like work by those that have not dealt with actually weighing a mouse. Bottom line—it is still too abstract for my tastes.

The discussion is better. But as mentioned above, female mice differ radically from male mice in their reproductive and behavioral goals. 10 or more litters over 10 months is an amazing feat of metabolism. Perhaps reading a bit of Kirkwood would help to embed your great data in the world of evolutionary theories of aging. His chapter in the 1990 book "Genetic Effects on Aging II" (ed. Harrison DE) is short and most interesting. It does not matter much that the females are virgins. What matters is their deep evolutionary history as wonderfully competent murine replicators. The results of this paper is relatively specific to the female state of metabolism and lifespan. The male life style is all about achieving polygynous success—about being the alpha mouse by 4-6 months of age. I am ok with the discussion as it is, but there is a missed opportunity to put this work into the context of the disposable soma theory. To me this would be even more interesting than the new text on candidate genes.

(Remarks on code availability)

Reviewer #4

(Remarks to the Author)

(Remarks on code availability)

Code looks good and will reproduce the analysis presented

Reviewer #5

(Remarks to the Author)

The authors present a novel analysis of body weight trajectories of genetically heterogeneous mice that were randomized to 5 different dietary restriction regimens. This is a noteworthy study and analysis of interest to readership in terms of methodology for analyzing healthspan, longevity, and genomics. The work contributes to the field by adding associations

between genetics and a type of bodyweight stability given stressors that are affected by diet. The analysis is thoughtful, and conclusions are well-supported by the analysis and presentation of data. The code and data were available and are clearly laid out (but this reviewer did not test these). An intriguing result is the analysis of measurement assays as a stressor on mice and how diet affects the probability of losing, gaining, or maintaining weight. The statistical methodology is very complex, and the explanations are good. The prior reviewers critiqued components of the analysis and results that were less clear, and the authors adequately responded and edited the manuscript and figures for clarity and provided additional analyses and results. There were two technical questions about the statistical methodology (ARHMM) that remain for me.

1. Figure 2B and 3E appear to suggest a pattern in transitions that are higher order such as a stressor recovery cycle, but this may be speculation on my part. Did the model assume time homogeneous transitions? Was this assessed? Is this a limitation?
2. Did the model consider the issues of survivor bias in assessing the analysis of the terminal decline of bodyweight? That is, mice that are declining are likely to die/drop out and create a nonrandom missing data problem. How could that affect results?

(Remarks on code availability)

The code is clearly written.

Version 2:

Reviewer comments:

Reviewer #5

(Remarks to the Author)

The authors have sufficiently responded to my concerns.

(Remarks on code availability)

REVIEWER COMMENTS

Reviewer #1 (Remarks to the Author):

In their manuscript, Prateek et al study the interaction between genetics and diet in determining lifespan. They split nearly one thousand outbred mice across 5 dietary regimes including various degrees of intermittent fasting and caloric restriction, and make daily measurements of lifespan from six months until 42 months of age. This is impressive data, both in its scope and in the quantitative quality of the data collected. The authors genotyped all animals using a microarray, which in combination with fully-sequenced inbred founders of the outbred population, allowed fine-grained evaluation of phenotype-genotype correlations.

The authors discretize their body-weight trajectories using a hidden Markov model that seems well suited for the task. They find, not unsurprisingly, that the degree of calorie restriction correlates with the time spent in various body weight states—gaining weight, stable weight, and losing weight. The authors then find that some aspects of body-weight dynamics during aging are associated with longer life, and identify genes associated with such body weight dynamics. The analysis appears rigorous but throughout the manuscript, the authors appear to deliberately avoid the most important question: what has this study taught us about the relationship between diet and lifespan? Remarkably, there is no heritability analysis of lifespan which naively one would suspect to be the main question being addressed by a study of this design.

In many places, the authors seem to be actively obscuring the true effect sizes of the effects of various diets on body weight dynamics and lifespan (see points 1 and 2 below), and in general, the authors can do a much better job clarifying for the reader the magnitude and quantitative importance of their findings.

Include other effect terms in fig. 4.: e.g. the direct effects of diet and body weight on lifespan PH ratios are useful, but what fraction of lifespan variance is explained by the heritable factors driving the dynamics of body weight? Fig 5b should show this, but is heritability in % or a fraction? How does this compare to the total heritability of lifespan across all loci (irrespective of their mediation via body weight dynamics)? Is the total heritability seen in this experiment similar to other estimates of heritability of lifespan in mice?

In the absence of a clear statement of the overall magnitude of the effect sizes being studied, the reviewer is left uncertain about how much concern should be placed on potential confounding factors. Across such a large cohort of animals, could confounding environmental factors be driving some of the associations identified?

We appreciate the reviewer's positive feedback regarding our work. We agree that the relationship between diet and lifespan, and the heritability of lifespan are important questions that this study was designed to address, and point the reviewer to Di Francesco *et al.* (now

published in *Nature*) for a comprehensive analysis of the impact of diet and genetics on lifespan for the same cohort of mice described in this work. We also point the reviewer to a more recent preprint by Mullis *et al.* that compares the heritability of lifespan in this cohort of mice with other cohorts.

To illustrate the effect sizes of body weight, diet, and interaction between diet and body weight, in the time-varying Cox proportional hazard, we have included a new supplementary figure in the revised manuscript (**Supplementary Figure 6**). Furthermore, in **Figure 5B** of the revised manuscript, we have illustrated the heritability of lifespan obtained from Di Francesco *et al.* We observed that the heritability of several of our body weight-derived phenotypes are comparable to that of lifespan.

All analyses, including the time-varying Cox proportional hazard model and the GxEMM model for estimating heritability and performing genome-wide association studies, incorporated generational waves as covariates to account for potential environmental confounding variables. Furthermore, in response to Reviewer 3, we quantified the proportion of variation in body weight explained by various factors in our study (age, diet, phenotyping assays, generation, cage, genetics) and found that environmental factors like generation and cage explain very little of the variation in body weight and our derived traits (**Supplementary Figure 1A**).

- Di Francesco, A., Deighan, A.G., Litichevskiy, L. *et al.* “Dietary restriction impacts health and lifespan of genetically diverse mice.” *Nature* 634, 684–692 (2024).
- Martin N. Mullis, Kevin M. Wright, Anil Raj, et al. “Analysis of lifespan across Diversity Outbred mouse studies identifies multiple longevity-associated loci.” bioRxiv 2024.11.20.624531.

Major points

1. The authors calculate statistics on body weight trajectories, for example, the longest continuous bout and the maximum rate of change of body weight. It's unclear how interpretable the statistics based on these extreme outliers are—for example, reasoning based on changes in “the average longest stretch of maintaining body weight” seems rather baroque. Are their simpler statistics, less likely to depend on the values of extreme outlier events, that can demonstrate the author's points? The reader is left wondering whether these exotic statistics were chosen to hide the fact that more straightforward statistics would leave the reader underwhelmed at the magnitude of the biological effects measured. Though the authors provide p-values for the statistical significance of these effects (in Fig. S3), it would probably be more helpful to readers to know the error in the estimate of the effect sizes, by providing confidence intervals.

We would like to clarify that we derived phenotypes based on both central tendency statistics (e.g., proportion of time spent in steady state) and extreme value statistics (e.g., longest bout in steady state), and reported results in both classes of phenotypes.

In **Supplementary Figure 4** (previously Supplementary Figure 3), we used the non-parametric Mann-Whitney U test due to its focus on data ranks rather than raw values and its robustness to outliers. While a Mann-Whitney U test does not inherently provide a confidence interval of the test statistic, an alternative approach is to report the area under the curve (AUC) along with the p-value. For a two-group classification problem, we can interpret the AUC as the probability of the classifier ranking a random observation from one group higher than a random observation from the other group.

- Hernández-Orallo, José, Peter Flach, and César Ferri Ramírez. "A unified view of performance metrics: Translating threshold choice into expected classification loss." *Journal of Machine Learning Research* 13 (2012): 2813-2869.

2. In Figure 4, the authors only provide the estimated hazard ratios for one arrow of their full model—the estimated direct effect of phenotype on lifespan. This leaves the reader unclear on the magnitude of these effects relative to other effects in the model, in particular the hazard ratios for diet's effect on lifespan and body weight's effect on lifespan. Can the authors include these direct effects somewhere prominently, to provide context for the phenotype->lifespan effects?

The authors thank the reviewer for their inquiry, as it prompted a revision of our estimation of the time-varying Cox proportional hazard model. Estimation of parameters of this model requires specification of a penalty which was previously fixed at $1e-5$. In our improved analysis, we use an empirical approach to specify this penalty. Specifically, splitting the set of mice into training and validation sets (80%:20%), we chose the penalizer term that yielded the highest concordance index (C-index) on the validation set. **Figures 4B, 4C, and 4D, and Supplementary Figures 7A, 7B, 7C, and 7D** have been updated accordingly. The revised manuscript now includes the following statement in the **Time-varying Cox proportional hazard model** subsection of **Methods** section:

Ln 407-409: We set the penalty term required by this function in an empirical manner. Specifically, we partitioned our set of mice into a training and validation set (80%:20% split) and performed a grid search to select the penalty value that yielded the highest concordance index (C-index) on the validation set.

Furthermore, we have also included a new supplementary figure to illustrate the effect sizes of body weight, diet, and the interaction between diet and body weight, on lifespan at six-month intervals (**Supplementary Figure 6**). As expected, the effect size of diet is larger than the effect size of the state occupancy in any latent state at all ages. Overall, the effect size of body weight is comparable in magnitude to the effect size of decline state occupancy on lifespan.

Supplementary Figure 6B: Effect sizes (+/- s.e.) of diet, and interaction between diet and body weight, for state occupancy in decline state as a function of age. Solid circles indicate p-values < 0.05, where the p-values were obtained from the estimated time-varying Cox model.

Similar patterns in effect sizes were observed for state occupancy in the steady and growth states (**Supplementary Figures 6C and 6D**). In the revised manuscript, we mention the following:

Ln 190-194: Across all three phenotypes, the effect size of 40% CR diet on lifespan was the largest. The effect sizes of body weight on lifespan were comparable in magnitude to the effect sizes of the phenotypes on lifespan. The effect sizes of the interaction between diet and body weight were significant only in the CR diets during late life (**Supplementary Figures 6B-6D**).

3. The presentation of the heritability of these traits is somewhat unclear to a non-expert reader. For example, this reviewer would have expected this paper would address the heritability of lifespan itself, which presumably will be low in outbred mice just as it is in human populations. Instead, the authors discuss the “heritability of traits associated with decreasing [or increasing] lifespan” which seems a bit of a confidence trick, talking about heritability and lifespan in the same figure heading but then presenting no data at all that quantifies the heritability of lifespan. This reviewer finds an analysis of the heritability of lifespan (which, for example, might be altered by the environment, with starvation increasing the influence of some SNPs on lifespan or decreasing the influence of others on lifespan) striking in its absence. Discussing the “heritability of traits” that are themselves “associated” with lifespan seems to be substituting a murky, indirect association in place of a much clearer analysis of the heritability of lifespan itself across the different conditions.

We acknowledge the confusion caused by our phrase “heritability of traits associated with decreasing [or increasing] lifespan”. We would like to clarify that, in **Figure 5B**, we have reported the heritability of body weight-derived traits; we have not reported the fraction of heritability of lifespan mediated by our derived traits. We have modified **Figures 5A and 5B** with clearer titles and subtitles, and have now highlighted the heritability of lifespan in **Figure 5B**.

We point the reviewer to Di Francesco *et al.* for a comprehensive analysis of heritability and genetic mapping of lifespan for the same cohort of mice described in this work.

- Di Francesco, A., Deighan, A.G., Litichevskiy, L. *et al.* “Dietary restriction impacts health and lifespan of genetically diverse mice.” *Nature* 634, 684–692 (2024).

We have included the following sentence, comparing the heritability of our derived traits with the heritability of body weight and that of lifespan:

Ln 224-227: Additionally, we observed that the heritability of the derived traits were comparable to the heritability of post-intervention body weight (genetics explained 20-30% of the variation in body weight) (Wright *et al.* (2022)) and the heritability of lifespan (genetics explained 23.6% of the variation in lifespan) (Di Francesco *et al.* (2024)).

4. Following up on the previous comment, the authors should make it clear what the overall heritability of lifespan is in this experiment. If lifespan has a very low or even 0 heritability in this experiment, this should be stated prominently.

We appreciate the reviewer's comment and have modified Figure 5B to illustrate the heritability of lifespan in this cohort of mice.

Figure 5B: Forest plot of the diet-specific and total (TO) heritability (+/- s.e.) of body-weight derived traits. (left panel): traits associated with decreasing lifespan; (right panel) traits associated with increasing lifespan. The dotted vertical line denotes the heritability of lifespan and the shaded region denotes +/- s.e. of heritability of lifespan.

5. The authors should include survival curves of some sort for all populations, to help readers and reviewers to estimate the quality of the underlying data. Presumably, the almost one thousand mice were not measured in a single age-synchronous cohort. Could batch-specific effects be influencing the findings and acting as a confounding factor in statistical analysis? For example, what is the effect size of the birth month on the time-dependent hazard rate?

We point the reviewer to Di Francesco *et al.*, where Kaplan-Meier survival curves of the mice enrolled in this study are shown (Fig 1b).

- Di Francesco, A., Deighan, A.G., Litichevskiy, L. *et al.* "Dietary restriction impacts health and lifespan of genetically diverse mice." *Nature* 634, 684–692 (2024).

The 960 mice in our study were enrolled in 12 generation waves, and these batches had little impact on the body weight trajectories and lifespan of mice (see our response to the comment on Generational Differences from Reviewer 2). Nevertheless, batch effects (generation wave) were controlled for in both our lifespan and genetic analyses.

Reviewer #1 (Remarks on code availability):

The code looks fine.

We thank the reviewer for reviewing our codebase.

Reviewer #2 (Remarks to the Author):

This paper investigates the impact of dietary interventions, specifically calorie restriction (CR) and intermittent fasting (IF), on body weight dynamics and lifespan in genetically diverse Diversity Outbred (DO) mice. Using advanced statistical models and gene mapping techniques, the study identifies key genetic loci associated with body weight regulation and longevity. The research provides valuable insights into the genetic and environmental factors influencing aging and metabolic health.

We appreciate the reviewer's recognition of the value of our findings in understanding the genetic and environmental factors influencing aging and metabolic health.

Comments and Suggestions

1. Generational Differences:

The study involves DO mice from different generations (22-24 and 26-28), which could result in body weight variation due to genetic and environmental factors specific to each generation. Given the current mouse housing and breeding setup, mice with earlier reproduction have a higher possibility to pass their genes to the next generation, which may also be associated with altered body weight.

Suggestion: Consider including generation as a variable in the analysis to assess its impact on body weight variation. Additionally, discussing potential reasons and implications of generational differences in body weight will help clarify if and how generational effects influence the observed outcomes and ensure the robustness of the study's conclusions.

The DO mice are maintained as a sizable colony of approximately 350 randomly assigned breeding pairs. At each generation, each pair passes on two offspring (one female and one male) to the next generation. This minimizes genetic drift and selection. Similarly, conditions in the mouse room where the current study took place remained stable throughout the entire study period.

We would like to clarify that generation waves were included as a covariate in both our lifespan and genetic analyses. Within each diet, no significant difference was observed in the average body weight trajectories between generations (see Figure below).

Supplementary Figure 1C: Average body weight trajectories of mice in each generation. Each panel corresponds to mice in different diets.

Furthermore, within each diet, there was no significant effect of generation on lifespan (see Figure below).

Response Figure: Kaplan–Meier survival curves stratified by generation within each diet. Significance was determined using multivariate log-rank tests comparing generations within each diet.

In the revised manuscript, we mention the following:

Ln 77-79: Within each diet group, we observed additional variation in body weight across mice and ages (Figure 1D); however generation (and associated confounders such as month of enrollment, season of the year, etc) contributed little to variation in the body weight trajectories (Supplementary Figure 1C).

2. Food Intake Adjustment:

The method of adjusting food amounts based on previous studies where “female DO mice at 6 months of age were estimated to consume 3.43 grams/day” is not scientifically rigorous. This is because food consumption in DO mice may vary significantly due to genetic diversity and may also vary with aging.

Suggestion: First, discuss the potential food consumption variation due to genetic differences and changes with age. Second, consider renaming the treatment categories; instead of using 20% and 40% food restriction, use terms like "mild" and "severe" restriction to better reflect the nature of the intervention.

We agree with the reviewer that food consumption in DO mice may vary significantly due to genetic diversity and aging. Our lack of precise measurements of food consumption as a function of mouse and age is indeed a key limitation of our study.

In our parallel study, Di Francesco *et al.* conducted a study of food consumption in a separate cohort of 160 diversity outbred mice under the same set of dietary interventions. Extended Data Figures 1a and 1b in Di Francesco *et al.* highlight the average cumulative food consumption of mice in each diet per day of the week and per week. These figures demonstrate that the average food consumption of mice in the AL group is very close to the amount used when designing the current study. Nevertheless, we acknowledge that food consumption in DO mice could be dependent on other factors including age (which these data do not address).

We have included the following paragraph in the Discussion, highlighting this limitation of our study.

Ln 308-317: One key limitation of this study is the lack of high resolution measurements of food consumption and energy expenditure in individual mice. We used an additional 160 DO mice,

subjected to the same dietary interventions at the same age, to evaluate daily and cumulative food intake (Di Francesco et al. (2024)). We observed that mice on IF diets lost body weight during the fasting period, displayed compensatory feeding after the fasting period, and recovered their body weight. In contrast, the CR mice had their food intake carefully regulated, by design. However, on weekends, these mice consumed their triple allotment of food by Saturday 15:00 (for 40% CR mice) and by Sunday 15:00 (for 20% CR mice). Despite this period of fasting, the body weight of CR mice showed little fluctuation over the weekend. Although there was high concordance between average food consumption and our study design at young ages, individual feeding behavior can change with age, and it is possible that some of the variability observed was due to varying degrees of individual caloric intake and expenditure.

We prefer to retain the names of the calorie restriction diets as 20% and 40% to be consistent with at least seven other papers (both published and in review) that use data from the same cohort of mice.

- Di Francesco, A., Deighan, A.G., Litichevskiy, L. *et al.* "Dietary restriction impacts health and lifespan of genetically diverse mice." *Nature* 634, 684–692 (2024).

3. Alternative Mathematical Models:

Since part of the readers of this paper will be gerontologists with a background in biology, it would be beneficial to discuss other potential mathematical models that can be used for determining the trend of body weight, besides the hidden Markov model used in this study.

Suggestion: Include a brief discussion of alternative models such as autoregressive integrated moving average (ARIMA) models, state-space models, and mixed-effects models. This comparison can provide additional context and support the robustness of the chosen methodology.

We appreciate the reviewer's suggestion to include a discussion of alternative models. While ARIMA, state-space, and mixed-effects models are valuable tools for time-series analysis, the autoregressive hidden Markov model (ARHMM) was chosen for its specific advantages in addressing the research question.

In the revised manuscript, we mention the following in the **Discussion** section:

Ln 258-262: In contrast, the ARHMM's ability to incorporate autoregressive components directly into the hidden state dynamics allows for capturing the temporal dependencies in body weight data, while the hidden states provide a framework for identifying distinct phases or regimes in body weight trajectories. This aligns with our goal of uncovering underlying patterns and shifts in body weight dynamics that may not be readily apparent from autoregressive integrated moving average (ARIMA) and mixed-effects models.

Overall, the authors provided a comprehensive and rigorous analysis of the effects of CR and IF on body weight dynamics and lifespan. The use of genetically diverse DO mice enhances the generalizability of the findings. The integration of advanced statistical models and gene mapping

techniques strengthens the validity of the conclusions. This study makes a significant contribution to the understanding of genetic and environmental influences on aging and metabolic health.

We thank the reviewer for the kind words regarding the rigor and significance of our findings.

Reviewer #2 (Remarks on code availability):

After a quick review of the provided code, it appears to be well-documented and potentially reproducible. The code includes a README file with clear instructions for installation and running the application, which enhances its usability for the community. Although I have not personally tested the code, the documentation and structure suggest that it should be repeatable. However, a thorough validation by running the code would be necessary to confirm this assessment fully.

We thank the reviewer for reviewing our code.

Reviewer #3 (Remarks to the Author):

Overview of Review. First, apologies in advance to the authors for some redundancy in these comments.

This is an excellent contribution to our understanding of the associations and causal linkages between body weight changes as a function of diet and four methods of dietary restriction. There is much gold here, but it takes a persistent reader or reviewer—one able to cycle through paper twice to recognize the merits of the work. This should be relatively straightforward problem to address. It is important however to optimize this paper so that it will have the impact it deserves. Many detailed suggestions below.

One of the major problems is that the introduction of this MS fail to highlight 1. the design, 2. the biological objectives, or 3. the main conclusions. Instead, the introduction focuses on what are relatively generic technical and computational issues. The technical and mathematical treatment is overwrought throughout this manuscript given the nature of the data.

Ideally, the introduction should also explain that this submission is a companion to at least two other papers widely circulating (and well received) on bioRxiv, including the Francesca paper cited by the authors.

Are the raw or processed data as shown in figure 1D available in supplements? The coverage of lifespan data is somewhat cryptic/hidden in this paper. We believe that the animals used here are same as those in both Francesco and Luciano papers. If true or if note, this needs to be stated explicitly.

We thank the reviewer for their comments regarding the impact and readability of our manuscript and are excited that they find our work an excellent contribution to the literature on diet, body weight, and lifespan. Yes, the animals used in our study and the data collected are exactly the same as those in the (now published) Di Francesco *et al.* paper. We have chosen to describe the design of the study briefly in our manuscript, while pointing the reader to the Di Francesco *et al.* paper for details. All the raw and processed data from this study, along with relevant code, are publicly available in the github repository (https://github.com/calico/do_bwd/). Below, we address in detail each point raised by the reviewer.

- Di Francesco, A., Deighan, A.G., Litichevskiy, L. *et al.* “Dietary restriction impacts health and lifespan of genetically diverse mice.” *Nature* 634, 684–692 (2024).

More important general (G) suggestions many regarding the introduction

G1. A very general comment on the submission: You want to ease readers into the biological importance and predictive utility of more complex derivatives of weight over lifespan as a function of diet and mild stressors at about 22 and 25 months. You do not want to confuse or impress readers with models that are actually very simple. Hold their hands. Guide them from simplicity to (modest) complexity. This is especially true if you want to be published in a general science journal like *Nature Communications* and to be cited heavily.

We have carefully revised the introduction to make it more accessible and easier for readers to understand the biological importance and predictive utility of derivatives of body weight on lifespan. Please see our revisions to the Introduction in our response to the reviewer's G2 comment. We believe that these changes will make the manuscript more suitable for the readers of a general science journal like *Nature Communications*.

G2. Along these lines: the introduction is not interesting, and it does not do justice to the data and interventions. Like the start of the discussion (more below), it is much too focused on technical and computational details. What hypotheses are you testing or what are you trying to predict (lifespan mainly, or GXE on lifespan)?

In the last paragraph of the introduction the words “measured”, “developed”, “characterized”, “demonstrated”, “explored” and “examined”, are used but most readers will want more than this—they will want and expect at least one or two new concepts and insights. This paper is not just about a fancier quantitative approach to parse and map higher moments of body weight. Biology, not methods and tweaks of Cox proportional hazards models. Although, I will add if this new method gets around the “proportional” assumptions is itself worth a paper.

Here are some concrete suggestions for a potentially stronger introduction. Introduce your design, the population, the actual broader context (this is one of at least three papers) and the questions you specifically address and answer. You do not need a long wind-up on the basics of aging and homeostasis. Focus on the baseline dynamics of body weight change and on the main levels of perturbations—diet and genetics—as predictors of lifespan. Deemphasize the

“high frequency” measures of body weight, which are at best only of moderate frequency. Some of us long-lived organisms weigh ourselves once or twice a day, and we will not be impressed by weighting once ever 10 days in no known phase with the cyclicity of four of the five diets. The “stress” mentioned a few times in this study is almost completely peripheral consequence of testing of the subset of animals that survived to 23 and 26 months. Be frank about this and explain the systematic weight losses at earlier ages.

Changes to the **Introduction** based on feedback from G1 and G2:

First paragraph:

Ln 18-20: Among the broad array of phenotypes typically measured in aging studies, body weight serves as a particularly accessible and integrative phenotype, reflecting the combined effects of genetics, behavior, environment, metabolic activity, and disease processes throughout life.

Third paragraph:

Ln 38-42: Using these time-series body weight data from 960 genetically diverse female mice, we derived novel phenotypes that go beyond static measures to describe dynamic processes such as weight stability, fluctuations, and recovery from stress. These phenotypes allowed us to ask specific questions: How do diet and genetics influence body weight homeostasis and dynamics as mice age? Do phenotypes derived from short-term fluctuations in body weight predict lifespan? How much does genetics influence these derived traits?

Fourth paragraph:

Ln 48-49: These findings highlight that dense longitudinal measurements and the resulting dynamics of body weight provide new insights into the biology of aging, revealing how genetics and environment shape health and longevity.

G3. The introduction does not introduce the key dietary manipulations or their motivations. Nor does it introduce the testing perturbations at 22 and 26 months. These topics are worth at least a full paragraph—diet and genotypes are instrumental variables, and the interventions at 22 and 26 months are a mild stressor applied when the mice are getting old.

We acknowledge our oversight in better describing the testing perturbations conducted in this study. We would like to clarify that each mouse in this study undergoes a battery of tests up to three times throughout its life (depending on how long it lives); not just at 22 and 26 months. We have now highlighted the dietary interventions in the introduction, and described the battery of tests both in the **Study design** subsection of the **Results** (**Ln 69-74**). While the set of tests remain the same each time, we have observed that the induced fluctuations in body weight depends on a number of factors including their age, diet, and body weight.

Figure 1C: Average body weight (solid lines) and standard errors (shaded region) of mice under different dietary interventions. Each horizontal gray bar represents a phenotyping event and the width of the gray bar captures the mean (+/-) standard deviation of the duration (in months) of the phenotyping event when 95% of mice were tested.

We modified **Figure 1C** to highlight the phenotyping schedules. Each horizontal gray bar represents a particular phenotyping event (detailed in **Supplementary Figure 1A**) and the width of the gray bar captures the time period during which 95% of mice have undergone the phenotyping event. The exact age of the mouse (in days) at the time of the phenotyping event is available in the file named “age_phenotype.csv” in the github repo associated with this manuscript.

In the revised manuscript, we mention the following:

Ln 69-74: Mice in the DRiDO study underwent three cycles of health assessments at early, middle, and late life. These assessments included a 7-day metabolic cage run at around 4, 15, and 27 months of age; blood collection for flow cytometry analysis at around 5, 17, and 29 months; rotarod, body composition, echocardiogram, acoustic startle, bladder function, free wheel running, and a blood collection for complete blood count at around 10, 22, and 34 months of age. Furthermore, manual frailty and grip strength assessments were carried out at 6 month intervals (Supplementary Figure 1A)

Ln 79-82: Some of the short-term variation in body weight could be attributed to the phenotyping tests to which the mice are subjected. For example, at 10, 24, and 35 months of age, all mice undergo nine tests over a period of one month, which involve handling by a technician or a change in their environment (Supplementary Figure 1B, Methods). Some of these tests induce stress, resulting in short-term fluctuations in body weight (Figure 1C).

G4. Since you only used females, this needs to be stated up front. This crucial “detail” should be in the abstract and also in the introduction. There are several good reasons for using only females. These statements and justifications must be stated sooner rather than later.

We have now highlighted, both in the abstract and the introduction, that only female mice were used in the study.

Abstract:

Ln 3-4: We measured body weight in 960 genetically diverse female mice, every 7-10 days over the full course of their lifespan.

Introduction:

Ln 29-31: To address these questions, we conducted a longitudinal study (DRiDO: Dietary Restriction in Diversity Outbred Mice) investigating the effects of dietary intervention on health and lifespan in genetically diverse female mice (Di Francesco et al. (2024)).

Ln 38-40: Using these time-series body weight data from 960 genetically diverse female mice, we derived novel phenotypes that go beyond static measures to describe dynamic processes such as weight stability, fluctuations, and recovery from stress.

Results:

Ln 52-55: We enrolled 960 diversity outbred (DO) female mice derived from eight inbred founder strains (A/J, C57BL/6J, 129S1/SvImJ, NOD/ShiLtJ, NZO/HILtJ, CAST/EiJ, PWK/PhJ, and WSB/EiJ) (Svenson et al. (2012)) across six generations of breeding, and measured their body weight every 7-10 days over the full course of their lifespan (Methods).

G4b. Much of the paper talks about homeostasis of body weight as if this were a goal in life. It is not, and this is particularly true of female mice when reproducing. So not only are you studying one sex, but it would be good to acknowledge that female mice are “built to produce offspring” and this is inherently a process that is associated with large fluctuations in the weight and metabolic state. I am not really criticizing the experimental paradigm, but I am saying that the fixation on “homeostasis” may be misplaced. What may be a better rubric is “metabolic resilience” or anti-fragility, and here the ability to maintain virgin female body weight may reflect resilience of the same type that a breeding female producing 5 litters over 8 months may need—even if her body weight oscillates at least over a range of 5 gm. Selection does not care about weight, it cares about fecundity and fitness.

We appreciate the reviewer’s comments on our choice of the term “homeostasis” vs “metabolic resilience”. As we elaborated in the **Discussion** section (see **Ln 279-289**), “resilience”, as usually used in the field of aging, is defined as the ability to recover (body weight) following an acute stress. Accordingly, mice recovering their body weight following periods of pregnancy would appropriately be referred to as “resilient”.

In this work, we used “homeostasis” to refer to maintenance of a particular physiological state (not limited to maintenance of body weight). In other words, a mouse steadily losing body weight over two months would be in a state of “decline homeostasis”. Additionally, in this work, we are referring to homeostasis over long periods of time (months), and our study is not designed to ascertain whether mechanisms (and evolutionary forces) that help maintain body weight resilience over multiple rounds of breeding are the same as those that maintain body weight homeostasis over the scale of lifespan.

G5. You have used state-of-the-art mapping methods, but it is not clear that you have identified “genetic determinants” of derivatives of body weight. The conventional alpha level for a genome-wide test is .05 and you have shifted the goal to 0.10. I can sympathize with this, but

others may be less sympathetic. In the methods there is nothing on the definition of support intervals, and at these modest linkage statistics they will be comparatively large.

In the revised manuscript, we now only report genomewide significant associations at the conventional alpha level of 0.05.

Therefore a bit mystified by the assertion that by fine-mapping you can get down to between 1 to 3 candidate genes. That means less than 100 Kb—beyond implausible with 954 mice divided into five very different diets with 64 haplotype combinations and with traits that are anything but Mendelian and with highly variable MAFs. Without a deep and very compelling dive into the support intervals, I would advise much more caution in claims of getting down to X to Y numbers of candidates for loci, and where there is a considerable risk that even small loci are the result of oligogenic variant "good luck" ascertainment bias. Look at what happened to Mackay's *Drosophila* lifespan loci.

We agree with the reviewer that we cannot assert that our approach of fine-mapping will get us 1 to 3 candidate genes; we have removed this sentence from the **Methods** section. We would like to clarify that we have described our fine-mapping approach in **Genetic fine-mapping with founder-allele-patterns** of the **Methods** section and have listed the support intervals identified by this approach in **Table 1** (column "FAP position"). In most cases, our support intervals span 100-700 Kb and cover 1-3 candidate genes.

G6. And on the same general topic—the current MS does not provide maps for the primary measurements—body weight itself (let alone any details on lifespan—I assume perhaps to avoid overlap with the Francesca paper?). The reader has no basis to determine if derivative measurements are truly adding all that much. I think that I am close to being convince, but it would be incredibly helpful (to the point of being essential) to provide the map for the primary measure of weights across ages and diets—perhaps even of the easy to understand weight changes per defined age interval. Also of interest to understand the correlation of body weight to its derivatives, prior to any "clean up".

Genetic maps for the primary measurement, body weight, were extensively reported in an earlier publication by Wright *et al.* and genetic map for lifespan, computed using this dataset, was reported in the Francesco *et al.* paper. While this manuscript was being revised, Mullis *et al.* published a preprint reporting a better powered genetic map for lifespan, using multiple datasets including the one from this study.

- Wright, Kevin M., et al. "Age and diet shape the genetic architecture of body weight in diversity outbred mice." *Elife* 11 (2022): e64329.
- Di Francesco, A., Deighan, A.G., Litichevskiy, L. et al. "Dietary restriction impacts health and lifespan of genetically diverse mice." *Nature* 634, 684–692 (2024).
- Martin N. Mullis, Kevin M. Wright, Anil Raj, et al. "Analysis of lifespan across Diversity Outbred mouse studies identifies multiple longevity-associated loci." *bioRxiv* 2024.11.20.624531.

Regarding the correlation of body weight-derived phenotypes with body weight, please refer to our response to comment R20 below.

G7. The last sentence of the introduction is problematic and includes the word “can” [and a typo]: “We expect that body weight measured at high temporal density can reveal patter[n]s of change that are associated with longevity and healthy aging.” Instead, you should ideally tell us something more positive and concrete—what are these patterns? What have you discovered?

We have changed the last paragraph of the **Introduction** based on the reviewer’s suggestion.

Ln 43-49: We observed that the ability to maintain body weight was associated with increased lifespan in an age-dependent manner. Furthermore, CR and IF diets appeared to train mice for broader stress resilience, extending beyond diet-induced stress, as evidenced by their sustained ability to recover from extrinsic stressors throughout their lifespan. Finally, we investigated the genetic contributions of body weight-derived traits associated with lifespan and identified several loci, none of which overlapped with genetic loci previously mapped to body weight itself (Wright et al. (2022)). These findings highlight that dense longitudinal measurements and the resulting dynamics of body weight provide new insights into the biology of aging, revealing how genetics and environment shape health and longevity.

G8. Figures 3 and 4 are the heart of the biology of this paper. In contrast, Figure 2, is a kind of geeky figure. It will be almost self-evident to most readers that the first derivative of body weight would have probably done the trick. In Figure 2B, the first example of a declared state transition to “loss” at 3.5 months, is surprising because weight is steady during this entire period and for a month before. (I realize this is just an example, but it is the first part of Figure 2B that anyone will look at closely.) The other “loss” states are correct. The question is why we have these first “loss” states occurred at all? One could suggest stochastic unhappiness of an individual mouse, but Figure 1D belies this idea since the drops in weight are seen across all diets at circa 11 and 15 months.

We have observed that the battery of tests that the mice undergo are an important contributor to short-term variation in body weight, consistent across all diets. In the example in **Figure 2B**, the first instance of state transition to “decline state” around 3.5 months is likely in response to this mouse spending a week in a metabolic cage. As illustrated in **Figure 2C**, our confidence in this inference is rather low, with the probability of “decline state” around 0.4, just slightly larger than the probability of each of the other two states. Our derived phenotypes account for the uncertainty in our state inference by weighting by the probability of the inferred states.

G9. I could see Figure 2 as being mildly useful in the supplement. Moving it out would enable room for one more figure focused on biology. Simple things, like body weight QTLs and KM functions and actual individual data across diets on the correlations between state occupancy and longevity. What would also be most interesting is the accuracy with which the mortality risk for an individual when they inevitably get close to the mortality cliff. This is evident in KM curves

and that is certainly prominent in the details of Figure 1D. I realize this is part of the Feb 8 bioRxiv preprint by Luciano et al. Just ping that paper.

We appreciate the reviewer's suggestion to replace **Figure 2** with one more focused on biology. We intended **Figure 2** to provide a visual description of how we modeled body weight trajectories throughout lifespan, an example of what our inferences look like against trends in the observed data, and our estimated model parameters. While we have described our model in detail in the **Methods** section (and the **Appendix**), we strongly believe that an illustration would give the readers of Nature Communications an intuitive understanding of our model, and, thus, we would like to retain **Figure 2**.

As mentioned in response to G6, body weight QTLs and Kaplan-Meier functions have been reported in Wright *et al.* and Di Francesco *et al.*, respectively.

- Wright, Kevin M., et al. "Age and diet shape the genetic architecture of body weight in diversity outbred mice." *Elife* 11 (2022): e64329.
- Di Francesco, A., Deighan, A.G., Litichevskiy, L. et al. "Dietary restriction impacts health and lifespan of genetically diverse mice." *Nature* 634, 684–692 (2024).

G10. Sorry, some redundancy in this comment. The temporal resolution of body weight data is not high enough to provide much in the way of bragging rights, especially since there is no claim to even pick up estrus cycle or diurnal effects. It boils down to three key states that anyone could pick out by eye or probably by taking $d(BW)/dt$. Do we really consider weights at 7 to 10 days high temporal resolution? Even daily weights are not high temporal resolution. Yes, I understand that this is 3 to 10 times higher than most previous work, but it turns out that it is not necessary for the main argument on stability of weight as a key predictor of lifespan. And the only true dynamics that were controlled at 22 and 25 months do not appear to play a prominent role in any of the results. However, this raises a potentially more interesting point on "homeostasis" versus anti-fragility. We metazoans need to be resilient, and a female mouse needs to be super-resilient during the first year of life when her fitness is tested most aggressively by all kinds of perturbations in the real world. This is why I disagree somewhat with the focus on homeostasis in this paper. The body weight of a wild non-virgin female mouse is anything but a regulated variable.

We consider our body weight measurements as high temporal resolution, particularly in relation to most aging studies (including ours) where phenotypes are typically collected at intervals of 6 months to 1 year (as highlighted by the reviewer). We agree that one could collect body weight at higher resolutions (e.g., daily / hourly); at these resolutions, we anticipate that our general ARHMM model could capture more than 3 states (for e.g., growth state could be further split into rapid growth and moderate growth), model nonlinear dynamics, and elucidate additional aspects of mouse physiology (such as estrus or diurnal effects). Please see our response to comment G4b, with regards to "homeostasis" vs "resilience".

G11. The current MS overstates some key genetic findings, and should be a bit more forthright about findings that are significant using standard criteria (95% confidence of association) and those that are interesting but require validation (< 95% confidence). Another issue is that the reader does not have access to information about loci for body weight done across time. As a result, the reader cannot be certain that the loci found for the derived traits would not also have been detected for variation in body weight at different time points during the experiment. The paper should go into more detail on how FDRs were computed, or if the FDRs are accounting for genome-wide significance only. There is also the issue that the first moment (body weight) has been processed so as to produce 18 derived (independent) traits for mapping. In theory, but rarely in practice, there could be a correction for the multiple derived phenotypes. A hat tip to this issue might be a classy addition.

We appreciate the reviewer's concern and have restricted **Table 1** to just the significant loci using standard criteria (95% confidence). As mentioned in response to G6, genetic loci associated with body weight across time were extensively reported in an earlier publication by *Wright et al.*. We found no overlap between our reported loci and those in the earlier publication, indicating that we are capturing distinct aspects of mouse physiology using the body-weight derived phenotypes. We have described in detail our approach to controlling for false discoveries in our genetic mapping (see **Genetic mapping with genotyped markers in Methods**). Indeed, we observed correlation between the derived phenotypes, and have specifically designed our FDR correction to account for this correlation.

- Wright, Kevin M., et al. "Age and diet shape the genetic architecture of body weight in diversity outbred mice." *Elife* 11 (2022): e64329.

G12. This really confused me when starting my review of the submission—the unexplained losses of weight losses at 11 and 15 months. This needs to be handled well in the paper. The weight losses due to stressful testing at 23 and 26 months will make good sense to the reader but they must needs to be marked on figure 1C (as done for the tests on the plots as in Figure 3E).

We would like to clarify that all mice in the study undergo a battery of tests, some of which occurred around 11 and 15 months of age (in addition to 23 and 26 months). Weight losses that consistently occur at these ages across all diets can be attributed to these testing events. We have now described the phenotyping events in the **Outline of study design** subsection of **Results** (see **Ln 69-74**), highlighted them in **Figure 1C** and **Supplementary Figure 1A**, and referred the reader to Di Francesco *et al.* for additional details on the battery of tests conducted.

- Di Francesco, A., Deighan, A.G., Litichevskiy, L. et al. "Dietary restriction impacts health and lifespan of genetically diverse mice." *Nature* 634, 684–692 (2024).

Results

R1 Minor. First sentence of results is either unneeded or must be documented. What is the error term of reweighing the same mouse every 10 minutes over an hour, or every hour over 24 hours? What level of accuracy and precision do you achieve? 0.5 g seems a reasonable goal.

We have removed the first sentence in the revised manuscript.

The scale used to weigh mice reports values to 0.1g, and it is calibrated for accuracy prior to each weighing session. The body weight of a mouse can fluctuate by up to 0.5g over the course of a day or week, depending on the type of dietary restriction intervention and time since last feeding. Body weights were taken on a specified day of the week for each group of mice, and the weighing period often spanned several hours, e.g., 8am to 12pm, during a period when mice are typically not feeding. Thus, a precision of +/- 0.1g is sufficient.

R2. Figure 1. Lovely overview of the design. If all cohorts that are part of the 960 DO sample were entered into the study during the same month or season that might be a worthwhile detail to add.

Each of the 12 cohorts of mice did not enter the study during the same month or season.

R3. How do you operationally define a month in days. This problem is trivial at one level but is also why some prefer days or weeks which are not as fuzzy in their definition.

In all our analyses, age was specified in days. To improve readability, throughout the manuscript, we have used months as the unit for age. Operationally, the age in days was divided by 30 and rounded to the nearest integer to obtain the age in months.

R4. Figure 1. Interesting that the 1-day/week fasting killed off an inordinately large number of females.

We are unsure where the reviewer has observed this in **Figure 1**. We did not observe a significant effect of one-day fasting on lifespan when compared to the ad-lib diet; refer to Kaplan-Meier survival curves for enrolled mice in Figure 1d of Di Francesco *et al.*

- Di Francesco, A., Deighan, A.G., Litichevskiy, L. *et al.* "Dietary restriction impacts health and lifespan of genetically diverse mice." *Nature* 634, 684–692 (2024).

R5. Fig 1A. Does "M" stand for mouse? This is a minor confusion since you did not use an males. If you mean numbers of mice just use the conventional symbol N. Not sure we need the mouse SVG, but what we definitely do need is the addition of the testing period shown in Figure 3E.

In the revised manuscript, we have used the variable N to denote the sample size in **Figure 1A**.

R6. Fig 1C. Please add tick marks for every month. What happened at 11 months and 23 months in the colony to cause such consistent drops in body weight across all cohorts? This is not normal and suggest a strong uncontrolled annual environmental perturbation. I would deduce that all cohorts maintained in one room.

In the revised manuscript, we have included ticks for every month on the x-axis in **Figure 1C**.

Between 11-15 and 22-26 months, mice undergo the first and second series of post-intervention phenotyping schedules, respectively. These phenotyping schedules include metabolic cages (MetCage), grip strength, frailty exams (including body temperature), blood draws for complete blood count (CBC), immune cell profiling by flow cytometry (FLOW), and fasted blood glucose measurements, home cage wheel running, voiding assay, rotarod, body composition by dual energy x-ray absorption (DEXA), echocardiogram (Echo), and acoustic startle (AS) (see **Supplementary Figure 1A**). These assays, differential handling by technicians, and transfer from group-housed to single-housed metabolic cages may induce stress in mice, which could explain some of the fluctuations in body weight in **Figure 1C**. We have now described these phenotyping events in the **Outline of study design** section (see **Ln 69-74**) of the revised manuscript. Conditions in the mouse room where the current study took place remained stable throughout the entire study period.

R7. Fig 1D. This is a critical figure and deserves much more room and contrast. The X-axis definitely needs an added “month-of-year” axis. The age-specific (or partly environment-season-specific) variance in body weights is what impresses me the most and there are clear wave patterns of changes across diets, most notable at roughly 11 and 14 months. 23 and 26 months. Were the diets initiated at precisely the same age and month? (Let’s see if this is discussed in terms of derivative measures.)

Mice were enrolled into the study at 3 weeks of age, in 12 cohorts (6 generations with 2 batches per generation); each cohort at different months of the year. Thus, including “month-of-year” information in **Figure 1D** would involve separating the traces into 12 groups, creating a crowded and visually unappealing figure. Instead, we have included **Supplementary Figure 1C** that shows the average body weight traces for each generation wave in each diet group. Details regarding the birth dates of mice enrolled in this study are available in the supplementary data of Di Francesco *et al.*

- Di Francesco, A., Deighan, A.G., Litichevskiy, L. *et al.* “Dietary restriction impacts health and lifespan of genetically diverse mice.” *Nature* 634, 684–692 (2024).

The fluctuations in body weight between 11-15 and 22-26 months can be partially attributed to the phenotyping schedules these mice underwent; refer to our response to comment R6. Diets were initiated at the same age (6 months); however, the diets were initiated at different months-of-year for each of the 12 cohorts.

Ideally many more individual traces would be resolvable. Death points need to be highlighted in some way (dots). The figure could benefit from an overlay of a KM-type survival estimator. Most females drop weight sharply before death. As one would expect the body weight drop is much reduced in the “40” group that do not have weight to spare, although it is easy to see the excessive deaths in this group at circa 23–24 months.

In **Figure 1D**, we have now included a dot to indicate the last time point at which body weight was measured before the mouse died.

R8. Fig 1D. In both the “20” and “40” cohorts a subset of females manages to become and maintain relatively obesity out to 24 months. Does this indicate that they are dominant females able to consistently obtain more food than their cage mates?

The mice in the 20% and 40% calorie restricted groups received a predetermined quantity of food daily, with each mouse provided a pre-weighed food pellet. However, for the weekend (Friday noon to Monday noon), three times the amount of food was placed in the cages of the CR mice, and it is possible that dominant females could consume more food consistently over the weekend. Although challenging to evaluate definitively, overt aggression was not readily apparent among the females. However, the possibility that food was not consumed equally within the shared environment cannot be excluded.

R9 Minor. P 4, line 45. The inbred founders of the DO need to be listed correctly. This nomenclature is wrong: AJ, B6, 129, NOD, NZO, CAST, PWK, and WSB. Please ensure RRIDs for founders and DO are listed in Methods.

A/J (AJ), Strain #000646, RRID:IMSR_JAX:000646
C57BL/6J (B6), Strain #000664, RRID:IMSR_JAX:000664
129S1/SvImJ (129), Strain #002448, RRID:IMSR_JAX:002448
NOD/ShiLtJ (NOD), Strain #001976, RRID:IMSR_JAX:001976
NZO/HILtJ (NZO), Strain #002105, RRID:IMSR_JAX:002105
CAST/EiJ (CAST), Strain #000928, RRID:IMSR_JAX:000928
PWK/PhJ (PWK), Strain #003715, RRID:IMSR_JAX:003715
WSB/EiJ (WSB), Strain #001145, RRID:IMSR_JAX:001145

In the revised manuscript, we listed the founder strains as follows:

Ln 330-332: These founder strains were A/J (RRID: 000646), C57BL/6J (RRID: 000664), 129S1/SvImJ (RRID: 002448), NOD/ShiLtJ (RRID: 001976), NZO/HILtJ (RRID: 002105), CAST/EiJ (RRID: 000928), PWK/PhJ (RRID: 003715), and WSB/EiJ (RRID: 001145).

R10. Minor. P 4, line 48. “At 6 months of age, randomly assigned to groups of mice were switched...” Delete “to”

Thank you for pointing out this error; we have now fixed this in the revised manuscript.

R11, P 4, line 61. Please express “largest” quantitatively (variance explained will do). The “substantial” other variation is a confound of age, season, and external environmental factors. At this point the careful reader has examined Fig 1 and has questions about the sources of variance that are not related to age or even diet.

We have now included a supplementary figure which captures the proportion of variance explained by the following covariates: age, cage, phenotyping assays, diet, generation, and mouse (see **Supplementary Figure 1B**).

Supplementary Figure 1B: Proportion of variance explained by various factors in the study.

R12, line 63. “Physiological”? What is the AR-HMM actually capturing—seasonality, testing, personnel changes, humidity fluctuations? All of these will of course have an impact on the physiology and anatomy of mice. Why should the reader assume hidden physiological state changes that are relatively uniform across all diets except the most extreme? Consider this comment a restatement of the puzzle of the weight losses earlier in life.

The ARHMM does not directly model the effect of seasonality, testing, personnel changes, or humidity fluctuations on the body weight of mice. Rather, it models underlying patterns in the dynamics of body weight that are shared across mice, irrespective of age and diet. Differences in these patterns were then, post-hoc, attributed to various factors in the study, including age, diet, and phenotyping tests.

We mention the following in **Modeling the dynamics of body weight** section:

Ln 86-88: These latent states correspond to distinct patterns in the body weight that are shared across all mice. They do not directly model the biological processes that determine a mouse's body weight (e.g., metabolic function, estrus cycle, activity levels, food consumption, etc).

R13, P 4, bottom paragraph. Many of your readers would benefit to an introduction to latent states—not as purely statistical entities but in the way a biologist will understand. You need to

motivate the abstractions and explain why simply taking a derivative of body weight by time or month or estrus cycle is not as useful. If latent states just translate to “gaining”, “stable”, “declining” then say so. If not, explain. Occupancy time in the state? Read Judea Pearl Book of Why if you need good examples of bringing statistical modeling concepts down to earth.

The latent states in our model are, indeed, statistical constructs that capture patterns in the dynamics of body weight that are shared across all mice in our study. These states do not directly model the biological processes that specify the body weight of a mouse at a given age (e.g., metabolic function, estrus cycle, activity levels, food consumption, etc). The states, estimated in our final model, are then labeled as “growth”, “steady”, and “decline” to link them to broad physiological processes.

R14, Fig 2B. This figure would also benefit from tick marks per month and a subsidiary X-axis that is month of year.

In the revised manuscript, we have included ticks for every month on the x-axis (see **Figures 2B and 2C**).

R15, P 6, lines 87-92. This is a wonderful text that belongs in the introduction. Autopoiesis is much more important than body weight homeostasis. In fact, body weight is a strongly modulated variable around an intentionally wide range in order to regulate metabolic function. ZJ Zhao and J Cao is just one example of work that may be relevant to some aspects of this submission (PMID:19447188). Hibernating species and many bat species take this to the extreme and also strongly modulate metabolic function daily or annually. Good data on non-commensal rodents such as *Peromyscus*. Entire books on restriction (see FH Bronson (1989), L Fishbein 1991).

We appreciate the reviewer’s comments regarding this paragraph. In this work, we do not directly measure or model homeostasis of body weight. Instead, we model the dynamics of body weight, and define the maintenance of constant body weight dynamics as new measures of homeostasis. Without introducing the reader to how we model body weight patterns, and the underlying latent states, we feel moving this paragraph to the introduction would be completely out of place.

R16, Fig 3E. Again, why so exceedingly stingy with x axis tick marks and labels. Two ticks for at 30+ month axes is beyond sparse.

In the revised manuscript, we have included ticks for every month on the x-axis (see **Figure 3E**).

R17, Fig 4A. This is more than a directed acyclic graph. It is an assertion of your model. Please put the indicator variables on top (diet). You also have 100K genotypes as potential indicator variables (perhaps put them in a cloud). Body weight, body weight dynamics (not just “phenotypes”), and other as candidate mediators, and lifespan is the outcome of interest. Since you have at least three states you regard as mediators that are explicitly plotted in B, C, and D.

Why not model them explicitly and why not test as an SEM? If you make these changes then Fig 4B, C, and D will make much more sense and may become a classic.

We thank the reviewer for their suggestion and have modified the title of **Figure 4A**. Our intent in **Figure 4** was to identify whether traits derived from body weight dynamics were predictive of lifespan. **Figure 4A** illustrates some of the potential confounders of an association between the derived traits and lifespan, specifically diet and body weight itself. Indeed, genetics could also confound our associations, although we do not account for this in our model. We agree with the reviewer that it would be great to estimate how much the body weight derived traits mediate the effect of diet on lifespan, using SEMs or mediation analyses. However, such mediation analysis is not our focus in **Figure 4**.

R18. Regarding figure 4A, you make an inference which is unclear to me. They state these derived phenotypes are lifespan-associated, but (early life) body weight itself is also a predictor of lifespan. The statement: "We estimated the age-dependent effect of our derived homeostatic traits on mortality hazard (Figure 4A)" conflicts however with figure 4a which shows the infographic of their assumption and model. In other words, figure 4A does not show the age-dependent effect of the derived homeostatic traits on mortality hazards, and it does not differentiate the effects of the derived traits relative to the age-dependent effect of body weight on mortality hazards.

We thank the reviewer for pointing out this inconsistency. In the revised manuscript, in **Figure 4A**, the "body weight" and "phenotype" nodes have been replaced with "body weight (at six-month intervals)" and "phenotype (at six-month intervals)" to denote that these measures are age-dependent.

R19, Figure 5a: Only five loci are found to be significant at the conventional $p < 0.05$ threshold, which means that there are 6 / 11 loci which are suggestively associated at $P < 0.10$. Furthermore, please make difference between the two associated traits on the X chromosome more obvious. It took several minutes to distinguish the two separate loci on X.

We have now revised our approach for genomewide control of false discovery rate given multiple, correlated body weight derived traits. With our updated genome-wide significance threshold at $\alpha = 0.05$, we identified 8 loci across 10 traits. The loci on chromosome X are no longer significant at the conventional genome-wide significant threshold.

R20. Supplemental Fig 6: How are the derived traits associated/correlated with the body weight? Please add the correlation to body weight to the figure. This correlation between the derived phenotypes and the body weight could be discussed in the results section, since now it is unclear if body weight itself would have captured the same effects.

We computed the partial correlation matrix between average body weight (at every six-month interval) and body weight-derived traits (at every six-month interval) after accounting for diet and generation. Averaging removes body weight dynamics when computing partial correlation, but

the body weight-derived trait represents these dynamics. In the figure below, both state occupancy in decline and growth states are associated with lower average body weight. While a negative correlation between state occupancy in a growth state and average body weight is logical, it is unclear if this negative correlation would also be present between state occupancy in a decline state and average body weight. Since we did not observe any broad patterns of association between body weight and its derived phenotypes, we opted to exclude this figure from the manuscript.

Response Figure: Partial correlation between body weight (at every six month interval bin) and body weight-derived trait. *** for $p < 0.001$, ** for $0.001 \leq p < 0.01$, and * for $0.01 \leq p < 0.05$.

Methods

M1. The Methods do not actually describe the process and frequency with which mice were weighed. The most detail I can find is on page 4, line 50: "...approximately once a week". No information on time of day, handling, taring, or precision and accuracy. Since two of the diets involve intermittent fasting days, and the caloric restriction involves a sort of binge feeding over the weekend, these details really matter, if and only if one insists on talking about high frequency longitudinal weight data acquisition.

Page 16, line 301 describes a data collating process and numbers of cases at 6-month intervals. Please refocus on the basics since this work will be read independently of other studies.

We agree with the reviewer that our description of the study design and data collection was rather limited. We have expanded on these details in the revised manuscript (see **Outline of study design** and **Mouse housing, feeding, and body weight measurements**).

M2. Six generations were sampled. There are five treatments. While there are 6 generational cohorts, the only cohorts that really matter much are the dietary treatments.

We would like to clarify that the mice in the study were sampled from 6 generations, with two batches in each generation. Mice in each batch (generation wave) were randomized to the five treatments; thus, all 12 cohorts include mice in all 5 treatments. We have now clarified this in the revised manuscript (see **Outline of study design** in **Results**).

M3. There is no text in the Methods on the calculation of support intervals, but there is a most improbable claim that the candidate intervals contain merely 1 to 3 genes. That implies a support interval of about 50 to 100 Kb—a miraculous level of precision using fewer than 954 highly recombinant DO mice treated using 5 very strong environmental perturbations.

We would like to clarify that our method for fine-mapping and calculation of support intervals is described in the Methods subsection **Genetic fine-mapping with founder-allele-patterns**. We agree with the reviewer that we cannot assert that our approach of fine-mapping will get us 1 to 3 candidate genes; we have removed this sentence from the **Methods** section. Please see our response to comment G5, which is related to this comment.

M4. On line 185 the authors promise to give more detail on their FDR correction in Methods. However, this part of the analysis discussed in the Method section, leaves more questions than it answers. To the reviewer it is unclear how it was done. e.g. “Expecting that the set of phenotypes in our genetic analyses were not all independent, we grouped the phenotypes into four groups to determine a group-specific genomewide-level significance threshold.”. Why four groups? How were the phenotypes grouped together? Why make a difference in which p values are selected in the group based on rejection of the null hypothesis (maximum p value, versus random p value). Furthermore, since there are four “independent” groups, why was the FDR applied to the α to the p value array of each group separately? Since we’re talking about body weight derived traits, should the FDR not correct across all four groups to minimize potential false positives?

In the revised manuscript, we have now described our revised, data-driven approach to determine the genome-wide significance threshold (see **Ln 429-441**). Specifically, we clustered phenotypes based on their pairwise genetic correlations and have now identified two major clusters (**Supplementary Figure 8**). We identified a cluster-specific genome-wide significance threshold at a false discovery rate of $\alpha = 0.05$, and chose the minimum of the significance thresholds between the two clusters as the p-value cutoff for FDR control across both clusters (see **Genetic mapping with genotyped markers** in the **Methods** section).

Discussion

D1. The first paragraph focuses almost exclusively on the modeling of body weight change and the relatively simple definitions of states—gaining, maintaining, and losing weight. What is far more interesting to biologists will be the impact of the diets and of the testing perturbations on body weight changes across the five diets and lifespan.

We believe we have made significant contributions in both methodological development (ARHMM) and the identification of novel findings (association of body-weight derived traits with lifespan, genetic determinants underpinning these traits). While we agree with the reviewer that the results of the impact of the diets and of the testing perturbations on body weight change are of interest, we feel it is equally important to discuss the methods that were employed, and their novelty and limitations.

D2. This sentence in the second paragraph is genuinely tantalizing: “However, no previous studies have quantified the effect of diet on body weight trajectories throughout life and how features derived from such trajectories are associated with lifespan.”

I have highlighted in italic what seems to be of greatest interest. But only one of the following sentences carries this idea forward, and without much added value: that longer occupancy in steady-state is associated with longer lifespan. Figure 4 is an indirect method of displaying this key result. There would be much more familiar ways of making this point, even something as conventional as survival curves by diet but more importantly by state occupancy. In other word, a plot with actual lifespan on the x axis to help readers “get it”.

In the revised manuscript, we have now included Kaplan-Meier curves for state occupancy in growth, steady, and decline states (**Supplementary Figure 6A**).

D3. The state changes are implicitly and sometimes even explicitly linked in this stressful life events. But there is to independent stressful life “event” but rather the imposition of a potentially stressful fixed change in the diet. The cyclicality of the diets is of course a strong potential cyclic stress that can be used as an independent variable. But here we expect daily weight data or at least three weightings per week to detect the cyclic body weight responses to dietary stress. Since the animals are group housed, conspecific dominance hierarchies are a highly likely source of stressful events, but these are not measured in the current study. The three states—steady, growth, decline cannot be ascribed to any independent variable in this study, although to this reader’s eye it looks like the main effect may be seasonal or vivarium related. The within-individual fluctuations in body weight are not exposed with any clarity.

We agree with the reviewer’s comment that we would need a much higher resolution of body weight measurements to capture the cyclic nature of food consumption, for the mice on fasting (weekly 1-2 day food deprivation) and for the mice on calorie restriction (triple feeding on weekends). Additionally, while analyses in this work, Di Francesco *et al.*, and Luciano *et al.* did not identify cage effects and dominance hierarchies as a major determinant of differential food consumption, further research with higher resolution measurements on food intake is needed to

resolve this question definitively. Finally, conditions in the mouse room where the current study took place remained stable throughout the entire study period.

Minor Comments on Discussion

D4. Minor. Given all we know about the temporal changes in body weight with age, it might help to clarify the benefits of 7–10 day intervals between weightings. Hourly over 24 h periods have clear biological significance. Your chosen intervals of 3 or 4 times per lunar month has potential biological significance (perhaps as a function of estrus). But most researchers would assume, perhaps incorrectly, that every 3 to 6 months is adequate. So the question is why you need this? I suspect most of us would just use a smoother over data at this high a resolution.

Mice were weighed weekly to ensure high-resolution lifetime body weight (BW) trajectories and to distribute the workload evenly among phenotyping staff. Additionally, mice were weighed, and coat color was recorded, at the time of each procedure to minimize and identify potential data entry errors.

While mice may be sensitive to lunar phases, even in windowless rooms, it is not expected to be a major factor. The estrus cycle in mice is three days long, and any effect on BW is likely to be less than fluctuations due to feeding. Therefore, any lunar effects would be attributed to noise in our analysis.

D5. Minor. Use of the word “healthspan” on p 2 line 16. There are now some good references in which “healthspan” has been operationalized in mice. The dynamic traits in this manuscript could become part of a non-static definition of healthspan. As always, there is some risk of circularity. For work on healthspan estimation in mice see the longitudinal work by D. Lamming and Ehninger.

We now cite Bellantuono et al. and Xie et al. in the revised manuscript.

Ln 15-18: Dense longitudinal phenotyping throughout the lifespan of an organism enables us to measure homeostasis by quantifying temporal relationships in one or more phenotypes at multiple time-scales, and associate these measures to healthspan (Bellantuono et al. (2020)) and lifespan ((Xie et al. (2022); Di Francesco et al. (2024))

- Bellantuono, I., de Cabo, R., Ehninger, D. *et al.* A toolbox for the longitudinal assessment of healthspan in aging mice. *Nat Protoc* **15**, 540–574 (2020).
- Xie, K., Fuchs, H., Scifo, E. *et al.* Deep phenotyping and lifetime trajectories reveal limited effects of longevity regulators on the aging process in C57BL/6J mice. *Nat Commun* **13**, 6830 (2022).
- Di Francesco, A., Deighan, A.G., Litichevskiy, L. *et al.* “Dietary restriction impacts health and lifespan of genetically diverse mice.” *Nature* **634**, 684–692 (2024).

D6. Minor. The qualifier “In general, non-invasive procedures” would apply to almost all mammals except mice and rats. Invasive and longitudinal procedures are precisely an advantage of rodent models in which genotypes can be replicated. Xie and Ehninger’s study in Nature Communications is an excellent example. The hybrid cross-sectional and longitudinal approach is also a strong advantage of DO, UM-HET3, and of course any set of inbred strains.

Based on the reviewer’s earlier feedback, we have made major changes to the Introduction and this qualifier is no longer discussed in the Introduction.

D7. Minor: p 2, line 19-20. Good spot to add some equivalent mouse studies; Xie et al. in particular.

Thank you for highlighting this; we have now cited Xie et al. as well. Please see our response to D5.

D8. Minor: p 2, line 23. Also cyclical by day, lunar month, years, and of course reproductive cycles. There is some work on life history and body weight fluctuations.

D9. Minor: p 2, line 25. This whole paragraph is bland to the point of being almost unreadable. Only the last sentence is crucial. Why not start with that as the topic sentence? The choice of references is amusing—citing papers from 2022 and 2014 as support for fundamental statements. A better selection for key papers and reviews on these topics would provide readers with more background on the importance of GxE. Perhaps cite SG Wright in addition to KM Wright since you are using derivatives of methods he developed. And one paper on diet (Yang et al. 2014) having an effect on body weight is an odd and arbitrary choice given the vast literature on this topic. The work of W. Atchley, Cheverud, Mackay, and a huge human literature seems more appropriate in this context.

In response to comments D8 and D9, along with comments G1 and G2, we have now revised the Introduction and included some of the recommended citations.

D10. Minor: p 2, line 35. Tense change from past to present (derived?)

Thank you; we have corrected this now.

Reviewer #3 (Remarks on code availability):

Question for Nature Communications editors

1. What about data availability for individual animals? I see a section on code availability, but no promises or hints regarding data (yet?). This is a serious issue given the this paper is submitted by a for-profit company that may do its best to try to maintain the data as proprietary. If the entire data set on lifespans and body weight are not provide, then I personally regard this as unpublishable.

We appreciate the reviewer's comments about data availability, and would like to clarify that all the raw and processed data related to this study, including individual level body weight, lifespan, and genotype data, have been made available as part of the github repository (https://github.com/calico/do_bwd). We value open science, and it is our policy to publish all data and code related to our research, whenever we submit a paper for publication.

2. What editors are handling the other two papers associated with these data in bioRxiv. Ideally, the review of these papers would be coordinated as some level. There will be at least three papers, and of the three that I have now read this is probably the least polished.

We appreciate the reviewer's comment and have made a sincere effort to improve the readability and presentation of our manuscript, based on the excellent comments and feedback from all reviewers. To clarify, there are currently five papers at various stages of review, across different journals, where data from this study contribute substantially.

Reviewer #4 (Remarks to the Author):

Reviewer #4 (Remarks on code availability):

Code quality is solid, helpful readme

We appreciate the reviewer's comment on the quality and accessibility of our code.

REVIEWER COMMENTS

Reviewer #2 (Remarks to the Author):

The revised manuscript by Prateek et al. investigates the effects of dietary interventions on body weight dynamics and lifespan in genetically diverse Diversity Outbred (DO) mice. I have carefully considered the authors' responses to the reviewers' comments, including mine.

The authors have addressed the main concerns raised during the initial review, providing useful clarifications regarding generational effects, variability in food intake, and the choice of statistical models. These additions enhance the transparency and interpretability of the work.

While a few minor points could still be refined, I believe the revised manuscript presents a more coherent and accessible account of the study. Notably, this work complements the recent Nature publication by Di Francesco et al. ("Dietary restriction impacts health and lifespan of genetically diverse mice," Nature 634, 684–692, 2024), which reports findings from the same mouse population. Together, these studies provide a broader view of how dietary interventions influence aging and metabolic outcomes in a genetically diverse context.

I'm in favor of publication and believe this paper will be intriguing to scientists in the relevant fields.

We are pleased that we were able to address the reviewer's comments. We appreciate the feedback provided, which has helped us enhance the quality of our manuscript.

Reviewers #3 and #4 (Remarks to the Author):

Much improved. This MS was solid in its first draft but it now covers the complex design and goals somewhat more effectively.

Abstract: First sentence a beast.

Introduction: Thanks for adding "female". But this is only needed once. What is needed is some consideration in discussion (optional as far as I am concerned) as to what your results imply about the metabolic demands on the sex that does all of the heavy lifting wrt reproduction.

"Enrolled" is not the right word for mice. Perhaps "enter" works better. But I like the idea of enrolling mice.

Greatly appreciate the addition of experimental design details at the start of the results and in Figure 1.

We appreciate the reviewers' acknowledgement that our response has enhanced the manuscript's overall quality. We have revised the sentence to incorporate the word 'enter' as recommended.

Ln 52-55: We entered a total of 960 diversity outbred (DO) female mice derived from eight inbred founder strains (A/J, C57BL/6J, 129S1/SvImJ, NOD/ShiLtJ, NZO/HILtJ, CAST/EiJ, PWK/PhJ, and WSB/EiJ) across six generations of breeding into the study, and measured their body weight every 7-10 days over the full course of their lifespan.

Results read well. My original plea was for a re-expression of higher order metrics on weight dynamics and linkage to lifespan that are parseable in terms of days per "whatever higher order unit" of weight you care to highlight. Reading this paper is a pleasure in some ways but a frustration in the sense that I do not have an intuition about effect sizes or ranges of differences in unit that most readers will be able to understand. Much hard core experimentation in this study but it reads like work by those that have not dealt with actually weighing a mouse. Bottom line—it is still too abstract for my tastes.

While we appreciate the reviewers' request to express effects in units of lifespan (e.g., days), it is not possible to directly convert log-hazard ratios into changes in survival time because the time-varying Cox model is semi-parametric: its coefficients represent relative changes in hazard, not changes in the time-to-event. Unlike a simple univariate, survival comparison, translating these effects on hazard from our multivariate model into effects on time would require simulation of survival curves under specific counterfactual scenarios (e.g., a 10% increase in steady-state occupancy), which in turn depend on the estimated baseline hazard and the joint covariate distribution. To keep interpretation clear and rigorous, we report the estimated effect sizes and, for intuition, include a worked example showing how a given coefficient maps to a percentage change in the instantaneous hazard within an age bin.

Ln 190-195: To provide a more intuitive interpretation of the effect size, we can translate the log proportional hazard ($\log(\text{PH})$) into a concrete percentage change in mortality risk. For instance, during the 6 to 12-month age period, the $\log(\text{PH})$ for the decline state is approximately +0.065. Since state occupancy is measured as a percentage, this indicates that for each one percentage point increase in the time a mouse spent in a state of weight loss, its instantaneous risk of death (hazard) increased by 6.7% ($\exp(0.065) \approx 1.0671$).

The discussion is better. But as mentioned above, female mice differ radically from male mice in their reproductive and behavioral goals. 10 or more litters over 10 months is an amazing feat of metabolism. Perhaps reading a bit of Kirkwood would help to embed your great data in the world of evolutionary theories of aging. His chapter in the 1990 book "Genetic Effects on Aging II" (ed. Harrison DE) is short and most interesting. It does not matter much that the females are virgins. What matters is their deep evolutionary history as wonderfully competent murine replicators. The results of this paper is relatively specific to the female state of metabolism and lifespan. The male lifestyle is all about achieving polygynous success—about being the alpha

mouse by 4-6 months of age. I am ok with the discussion as it is, but there is a missed opportunity to put this work into the context of the disposable soma theory. To me this would be even more interesting than the new text on candidate genes.

We have now added a paragraph to our Discussion section, to put our observations in this work into the context of the disposable soma theory.

Ln 294-306: Our findings on the unique metabolic and lifespan characteristics of female mice can be contextualized within the evolutionary framework of the disposable soma theory of aging (Kirkwood and Holliday (1979)). This theory posits an inescapable trade-off between an organism's investment in reproduction versus the maintenance and repair of its own body. The female mouse, in particular, has an evolutionary history shaped by the metabolic demands of reproduction, requiring a physiology optimized for high energy turnover (Seli et al. (2014)). Our results align with the consequences of such a trade-off. We can interpret the steady state, which is strongly associated with increased lifespan, as a period of successful somatic maintenance. Conversely, the decline state, which predicts mortality, represents the failure of this maintenance as the soma becomes disposable. The observation that caloric restriction dramatically increases time spent in the protective steady state (Selesniemi et al. (2008)) further supports this framework; CR acts as a signal to defer costly reproduction and reallocate resources toward survival and somatic repair. Although the females in our study were virgins, their biology is a product of these deep evolutionary pressures, meaning the trade-offs described by the disposable soma theory are intrinsically active in shaping their physiology and aging trajectory. This may provide an ultimate explanation for why the dynamics of body weight homeostasis are so tightly linked to longevity in female mice.

- Kirkwood, T. B. L. and Holliday, R. (1979). The evolution of ageing and longevity. *Proceedings of the Royal Society of London. Series B. Biological Sciences*, 205(1161):531–546.535
- Seli, E., Babayev, E., Collins, S. C., Nemeth, G., and Horvath, T. L. (2014). Minireview: Metabolism of female reproduction: Regulatory mechanisms and clinical implications. *Molecular Endocrinology*, 28(6):790–804.566
- Selesniemi, K., Lee, H.-J., and Tilly, J. L. (2008). Moderate caloric restriction initiated in rodents during adulthood sustains function of the female reproductive axis into advanced chronological age. *Aging Cell*, 7(5):622–629.564

Reviewer #5 (Remarks to the Author):

The authors present a novel analysis of body weight trajectories of genetically heterogeneous mice that were randomized to 5 different dietary restriction regimens. This is a noteworthy study and analysis of interest to readership in terms of methodology for analyzing healthspan, longevity, and genomics. The work contributes to the field by adding associations between genetics and a type of bodyweight stability given stressors that are affected by diet. The analysis is thoughtful, and conclusions are well-supported by the analysis and presentation of

data. The code and data were available and are clearly laid out (but this reviewer did not test these). An intriguing result is the analysis of measurement assays as a stressor on mice and how diet affects the probability of losing, gaining, or maintaining weight. The statistical methodology is very complex, and the explanations are good. The prior reviewers critiqued components of the analysis and results that were less clear, and the authors adequately responded and edited the manuscript and figures for clarity and provided additional analyses and results. There were two technical questions about the statistical methodology (ARHMM) that remain for me.

We appreciate the reviewer's positive feedback regarding our work.

1. Figure 2B and 3E appear to suggest a pattern in transitions that are higher order such as a stressor recovery cycle, but this may be speculation on my part. Did the model assume time homogeneous transitions? Was this assessed? Is this a limitation?

We thank the reviewer for this insightful question. The reviewer is correct to point out that the transitions between physiological states are likely time-inhomogeneous and are influenced by events such as aging and phenotyping stress. The ARHMM we employed assumes time-homogeneous transitions. We made this assumption to first build a robust and interpretable baseline model of the fundamental physiological states (decline, steady, growth) based on the body weight dynamics alone. This allows us to first learn what these states represent in a general sense, and then analyze post-hoc how they are perturbed.

While using a more complex, time-inhomogeneous ARHMM is a potential alternative, we believe learning such a model would require substantially denser sampling of measurements and more animals in order to adequately capture time-inhomogeneous transition rates.

2. Did the model consider the issues of survivor bias in assessing the analysis of the terminal decline of bodyweight? That is, mice that are declining are likely to die/drop out and create a nonrandom missing data problem. How could that affect results?

We are unclear to which analysis the reviewer is referring. For analysis reported in Figure 3A, we agree with the reviewer that survivor bias likely influences some of observations in the last two age-bins. However, when the axes are scaled to 'proportion of life lived' as in Figure 3B, the terminal decline of body weight is not influenced by survivor bias since all mice contribute to observations in the last decile of life.

For our lifespan analyses, we employed a time-varying Cox proportional hazards model, a survival analysis framework robust to the potential confounding effects of survivor bias inherent in longitudinal aging studies. These models are designed to handle time-to-event data where subjects drop out due to death. The data from mice that die early (including those undergoing a severe terminal decline) informs the model's estimation of hazard risk and is not treated as nonrandom missing data. We believe our core conclusions about the association between body weight dynamics and lifespan are robust to the effects of survivor bias.